# Dwarf open reading frame (DWORF) is a direct activator of the sarcoplasmic reticulum calcium pump SERCA

M'Lynn E Fisher[1], Elisa Bovo[2†], Rodrigo Aguayo-Ortiz[3†], Ellen E Cho[2], Marsha P Pribadi[2], Michael P Dalton[2], Nishadh Rathod[1], M Joanne Lemieux[1], L Michel Espinoza-Fonseca[3], Seth L Robia[2], Aleksey V Zima[2], Howard S Young[1*]

[1]Department of Biochemistry, University of Alberta, Edmonton, Canada; [2]Department of Cell and Molecular Physiology, Stritch School of Medicine, Loyola University Chicago, Maywood, United States; [3]Center for Arrhythmia Research, Department of Internal Medicine, Division of Cardiovascular Medicine, University of Michigan, Ann Arbor, United States

**Abstract** The sarco-plasmic reticulum calcium pump (SERCA) plays a critical role in the contraction-relaxation cycle of muscle. In cardiac muscle, SERCA is regulated by the inhibitor phospholamban. A new regulator, dwarf open reading frame (DWORF), has been reported to displace phospholamban from SERCA. Here, we show that DWORF is a direct activator of SERCA, increasing its turnover rate in the absence of phospholamban. Measurement of in-cell calcium dynamics supports this observation and demonstrates that DWORF increases SERCA-dependent calcium reuptake. These functional observations reveal opposing effects of DWORF activation and phospholamban inhibition of SERCA. To gain mechanistic insight into SERCA activation, fluorescence resonance energy transfer experiments revealed that DWORF has a higher affinity for SERCA in the presence of calcium. Molecular modeling and molecular dynamics simulations provide a model for DWORF activation of SERCA, where DWORF modulates the membrane bilayer and stabilizes the conformations of SERCA that predominate during elevated cytosolic calcium.

*For correspondence:
hyoung@ualberta.ca

†These authors contributed equally to this work

## Introduction

The sarco-endoplasmic reticulum calcium pump (SERCA) is an ion transporting ATPase that plays a critical role in intracellular calcium signaling. SERCA maintains calcium content of the sarco-endoplasmic reticulum, creating a 2000-fold gradient that can be mobilized for signaling via inositol-1,4,5-triphosphate receptor (IP$_3$R) and ryanodine receptor (RyR) calcium channels. This signaling mechanism is essential for normal cell physiology, and disordered calcium handling underlies a diverse array of diseases including cardiomyopathies (*MacLennan and Kranias, 2003*), skeletal muscle disorders (*Treves et al., 2017*), and neurological diseases (*Mekahli et al., 2011*). In the heart, the cardiac-specific isoform SERCA2a maintains sarcoplasmic reticulum (SR) calcium stores, which is the major source of calcium for cardiac contractility. While SERCA calcium transport is essential for all cells, it is particularly important in cardiomyocytes where defects in SERCA activity or regulation are associated with cardiomyopathies and heart failure (*Schmitt et al., 2003*; *Haghighi et al., 2006*; *Haghighi et al., 2003*). This connection with heart failure has focused attention on SERCA as a possible target for therapy. An initially promising approach in animal models, increasing SERCA expression via gene delivery (*Byrne et al., 2008*), has proven challenging in human clinical trials (*Hulot et al., 2017*). Nonetheless, the rationale for treating heart failure by improving calcium transport function remains compelling and orthogonal approaches may be required.

In the heart, SERCA activity is regulated to allow dynamic calcium homeostasis, which changes in response to the need for cardiac output during rest or exertion. A small transmembrane protein, phospholamban (PLN), is the primary regulatory subunit of SERCA in cardiac muscle. PLN inhibits SERCA by decreasing its apparent calcium affinity, thereby reducing both the rate of calcium reuptake and the amount of calcium in the SR. PLN inhibition of SERCA controls the rate of relaxation and the SR calcium load for subsequent contractions, modulating both the lusitropic and inotropic properties of the heart. PLN inhibition of SERCA is relieved by phosphorylation with the main mechanism involving protein kinase A (PKA) and the β-adrenergic pathway (*Kirchberber et al., 1975*). Under resting conditions, PLN is an important brake that prevents SR calcium overload and the arrhythmogenic consequences (*Desantiago et al., 2008*). In turn, regulation of PLN by phosphorylation – regulation of the regulator – creates a dynamic SR calcium load that can respond to the sympathetic need for increased cardiac output, the so-called 'fight-or-flight' response. Thus, there exists a pool of SERCA pumps in the SR membrane and their combined calcium transport capacity can be finely tuned – upregulated or downregulated – depending on the requirement for cardiac contractility.

Since the initial discovery of PLN decades ago (*Kirchberber et al., 1975*), it has stood as the only regulatory subunit of SERCA in ventricular muscle. This changed recently with the discovery of a small transmembrane SERCA-binding protein, dwarf open reading frame (DWORF) (*Nelson et al., 2016*). DWORF has weak sequence similarity to PLN and its skeletal muscle homolog sarcolipin (SLN), and it appears to comprise a transmembrane peptide of unknown structure and function (*Figure 1*). DWORF was found to increase SERCA activity by opposing PLN function, leading to the hypothesis that DWORF is a non-inhibitory competitor that displaces PLN from the inhibitory groove of SERCA (*Makarewich et al., 2018*). This raised the question, why is an additional means of reversing PLN inhibition necessary? PLN inhibition can be reversed by phosphorylation (*Karczewski et al., 1997*; *Catalucci et al., 2009*), elevated calcium concentrations, and we can now add DWORF as another level of redundancy in reversing SERCA inhibition by PLN. In the present study, we show that DWORF is a direct activator of SERCA activity, in addition to its role in displacing PLN from the inhibitory groove of SERCA. To elucidate DWORF function, we compared the ability of DWORF and PLN to directly regulate SERCA function. We measured the effects of DWORF on SERCA in a membrane reconstitution system, and we measured SERCA-dependent calcium dynamics and SERCA binding by DWORF during active intracellular calcium signaling in cells. We then modeled the structure of DWORF and the SERCA-DWORF complex, and evaluated their properties using molecular dynamics (MD) simulations. The results provide insight into DWORF structure and function and a role for DWORF in the direct regulation of the SERCA calcium pump.

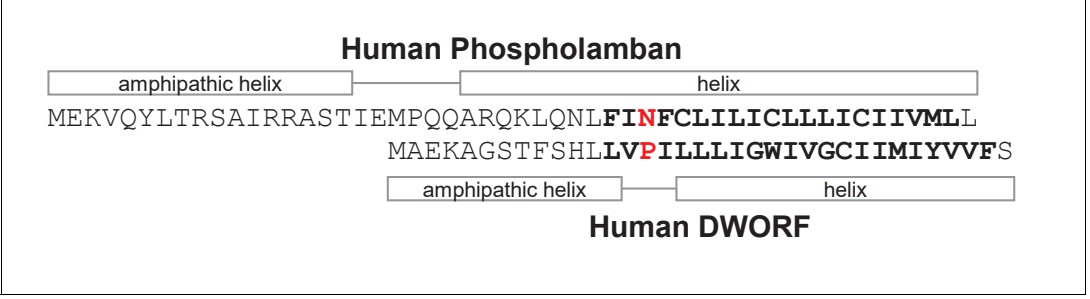

**Figure 1.** Sequence alignment, secondary structure prediction, and transmembrane domain prediction for phospholamban and dwarf open reading frame (DWORF). The predicted transmembrane domains are in bold letters. The sequences are aligned between Asn[34] of phospholamban and Pro[15] of DWORF (*Primeau et al., 2018*). The helical regions of phospholamban are based on the NMR structure (PDB code 2KYV). The helical regions of DWORF are based on sequence predictions and molecular dynamics simulations in the present study.

## Results

### SERCA activity in the presence of DWORF

Previous studies of DWORF function focused on the apparent calcium affinity and not the maximal activity of SERCA in the presence of DWORF (*Nelson et al., 2016*; *Makarewich et al., 2018*). These studies demonstrated that DWORF does not alter the apparent calcium affinity of SERCA. A recent study claimed to show that DWORF increases the apparent calcium affinity of SERCA (*Li et al., 2021*), though our current data (*Figure 2*) and previous studies by others (*Nelson et al., 2016*; *Makarewich et al., 2018*) do not support their conclusion. To investigate if there was a direct functional effect of DWORF on the apparent calcium affinity ($K_{Ca}$) or maximal activity ($V_{max}$) of SERCA, we used an in vitro membrane co-reconstitution system (e.g., *Ceholski et al., 2012*; *Trieber et al., 2009*; *Trieber et al., 2005*). To achieve this, DWORF was co-reconstituted into membrane vesicles with SERCA and the calcium-dependent ATPase activity was measured (*Figure 2*). The previously

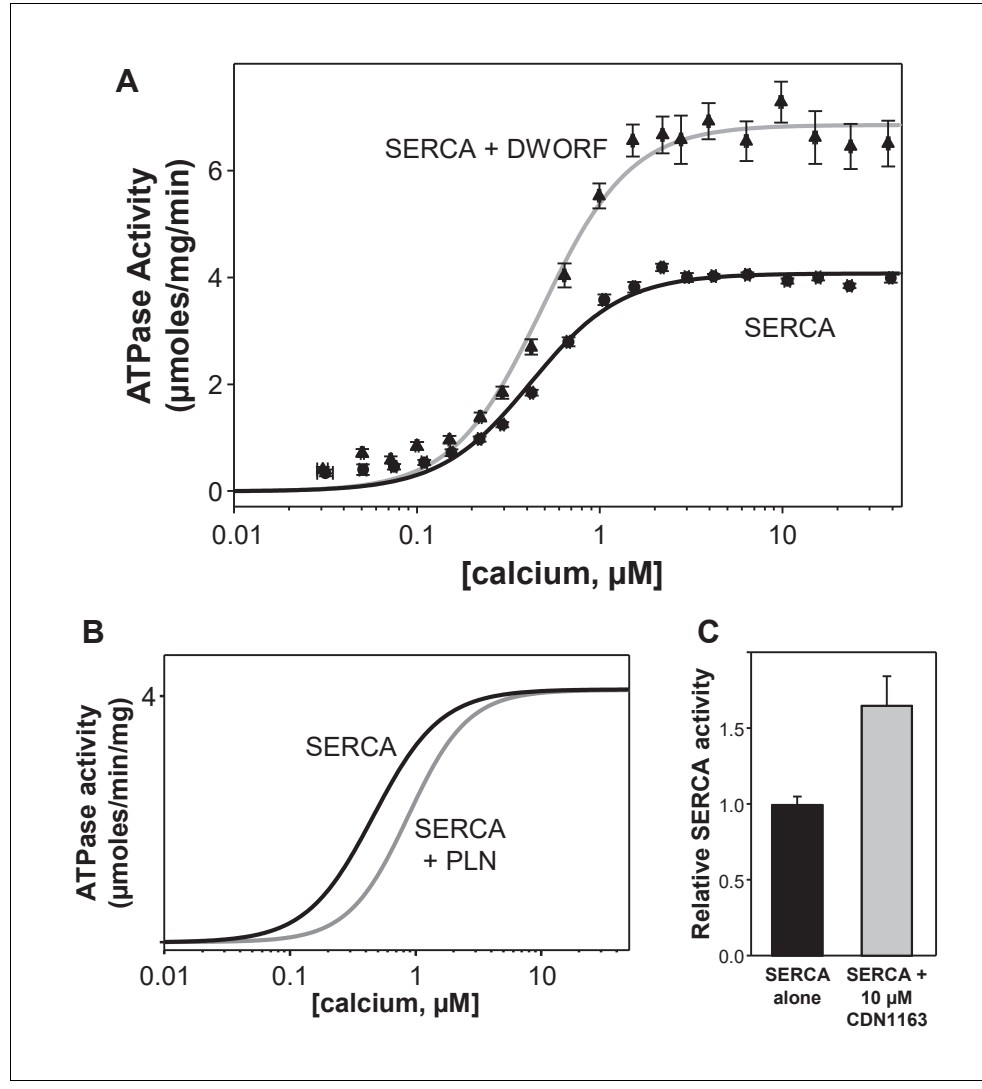

**Figure 2.** ATPase activity of sarco-endoplasmic reticulum calcium pump (SERCA) co-reconstituted into membrane vesicles in the absence and presence of dwarf open reading frame (DWORF). (**A**) ATPase activity of SERCA membrane vesicles in the absence (black line) and presence of DWORF (gray line). (**B**) Comparison to previously published ATPase activity for SERCA in the absence (black line) and presence of phospholamban (gray line). Only the fitted curves from reference (*Anderson et al., 2015*) are shown. (**C**) ATPase activity of SERCA membrane vesicles in the absence (black bar) and presence (gray bar) of the small molecule activator CDN1163 at saturating calcium concentration (3 µM).

reported ATPase activity curves for SERCA in the absence and presence of PLN (*Glaves et al., 2019*; *Gorski et al., 2013*) were compared to the activity curves for SERCA in the absence and presence of DWORF. At a molar ratio of ~1–2 DWORF per SERCA, DWORF increased the turnover rate of SERCA at nearly all calcium concentrations tested (0.01–15 µM free calcium). The maximal activity ($V_{max}$) of SERCA in the absence of DWORF increased 1.7-fold in the presence of DWORF ($V_{max}$ values were $4.1 \pm 0.$ and $6.9 \pm 0.1$ µmol/min/mg, respectively). This level of SERCA activation is comparable to the small-molecule activator CDN1163 (*Kang et al., 2016*). There was no significant effect of DWORF on the apparent calcium affinity of SERCA ($K_{Ca}$ values of $0.42 \pm 0.03$ and $0.48 \pm 0.03$ µM calcium, respectively). Thus, DWORF acted as a direct activator of SERCA by increasing its maximal activity (turnover rate).

## Cellular calcium transport in the presence of DWORF

Previous studies have shown that DWORF relieves SERCA inhibition by displacing PLN from the inhibitory groove of SERCA (*Nelson et al., 2016*; *Makarewich et al., 2018*). Our data are consistent with this observation, and we add a new facet of DWORF function as a direct activator of SERCA even in the absence of PLN. To determine whether increased ATPase activity was accompanied by increased cellular calcium transport activity, we utilized a newly developed approach for measuring calcium uptake into the endoplasmic reticulum (ER) of live cells (*Bovo et al., 2019*). The technique used an inducible human SERCA2a stable cell line (t-Rex-293 cells) containing the calcium release channel RyR2, which allows manipulation of ER calcium load, and an ER-targeted calcium indicator R-CEPIA1er, which allows measurement of ER calcium load (*Bovo et al., 2019*; *Bovo et al., 2016*). The cell line was transiently transfected with mCer-DWORF (or mCer-PLN) to assess regulation of SERCA2a. mCer-DWORF was predominantly expressed in the ER (estimated from the overlap between the mCer-DWORF and the R-CEPIA1er signals; *Figure 3A*), with a distribution pattern similar to SERCA (*Bovo et al., 2019*). To determine SERCA2a function, the plasma membrane was permeabilized with saponin to allow control of the cytosolic-free calcium and ATP concentrations. ER calcium uptake was quantified from the increase in R-CEPIA1er fluorescence after full ER calcium depletion by caffeine followed by RyR2 inhibition with ruthenium red and tetracaine (RyR + Tetr; *Figure 3B*), which blocks the principal calcium leak pathway (*Bovo et al., 2019*). The first derivative of ER calcium uptake ($d[Ca^{2+}]_{ER}/dt$) was plotted against the corresponding $[Ca^{2+}]_{ER}$ to estimate the maximum ER calcium uptake rate and the maximum ER calcium load (*Figure 3C*). As expected, PLN transfection decreased ER calcium uptake compared to control. The opposite effect was observed with DWORF transfection, which almost doubled SERCA calcium uptake rate over the entire range of physiological ER calcium loads. These data indicated that DWORF acts as a potent activator of SERCA2a and this regulation is direct, not requiring pre-existing inhibition of SERCA by PLN. Moreover, it was noteworthy that DWORF enhanced the maximum ER calcium load (*Figure 3C*, arrow). Since the thermodynamic driving force for calcium transport is the ATP/ADP ratio, we conclude that DWORF enhances SERCA catalytic efficiency and reduces the energetic cost of calcium transport (*Bovo et al., 2019*).

To investigate the effect of DWORF on SERCA in cellular calcium handling dynamics, we mimicked cardiac calcium handling in the heterologous cell model expressing SERCA and RyR (*Figure 4*, *Figure 4—figure supplement 1*). This system generates periodic calcium waves due to spontaneous calcium release by RyR2 followed by calcium reuptake by SERCA (*Figure 4A, B*). Co-expression of DWORF resulted in increased ER calcium load (*Figure 4B*; $[Ca^{2+}]_{ER}$) and increased frequency (*Figure 4B*; Tau(s)) and amplitude of spontaneous calcium waves, suggesting a significantly faster calcium reuptake due to increased SERCA activity. In contrast, PLN significantly decreased the calcium uptake rate by slowing the calcium wave recovery. Similar effects of DWORF on spontaneous calcium release were observed when calcium waves were measured as cytosolic calcium fluctuations in intact cells (*Figure 4C*) with a significant increase in the average amplitude and integral of RyR-mediated calcium release events in DWORF-expressing cells (*Figure 4D*). These results demonstrate that DWORF expression increases ER calcium load by enhancing SERCA-mediated calcium uptake. These data support the observation that DWORF directly increases the calcium-dependent ATPase activity of SERCA (*Figure 2*).

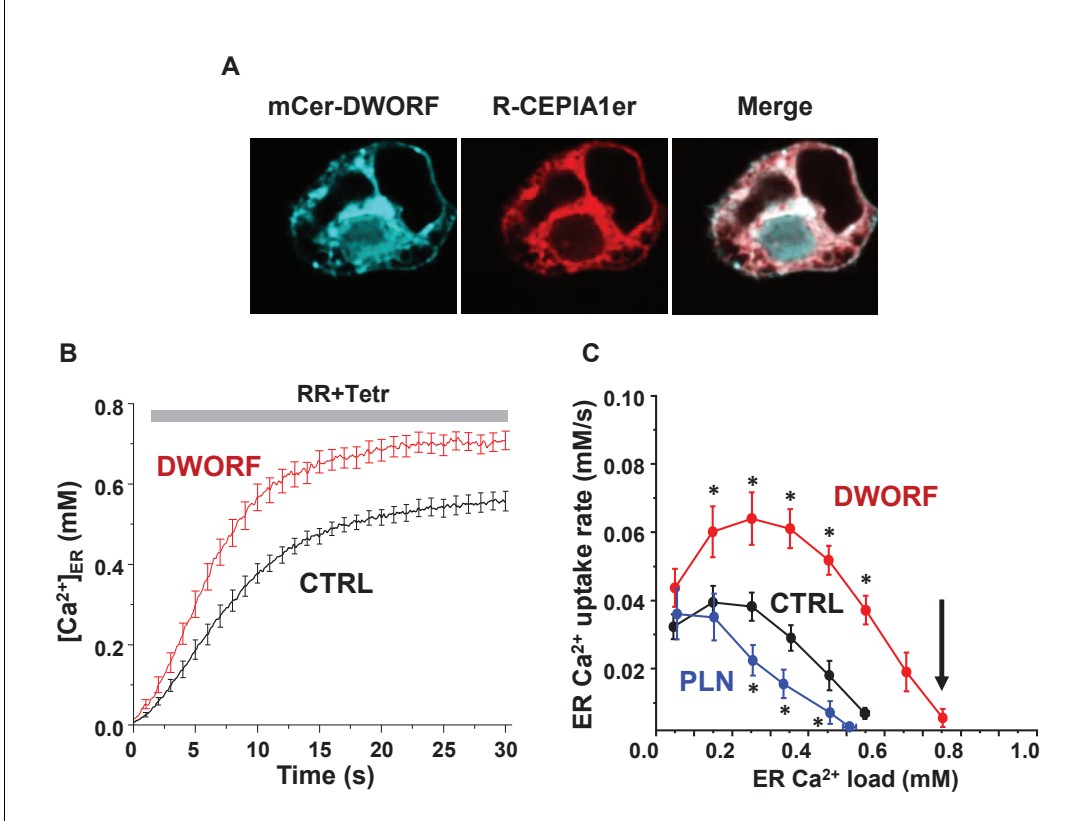

**Figure 3.** Dwarf open reading frame (DWORF) enhances sarco-endoplasmic reticulum calcium pump (SERCA)-dependent calcium dynamics in cells. (A) Inducible human SERCA2a stable cell line (t-Rex-293 cells + Tetr) was transfected with mCer-DWORF together with the $Ca^{2+}$ release channel ryanodine receptor (RyR)2 and the endoplasmic reticulum (ER)-targeted $Ca^{2+}$ indicator R-CEPIA1er. MCer-DWORF expression in these cells showed a similar pattern as R-CEPIA1er. (B) The rate of $[Ca^{2+}]_{ER}$ reuptake after full ER $Ca^{2+}$ depletion by caffeine followed by RyR2 inhibition with ruthenium red (RR). (C) The ER $Ca^{2+}$ uptake rate was plotted against the corresponding ER $[Ca^{2+}]$ load. DWORF expression almost doubled ER $Ca^{2+}$ uptake rate through the entire range of physiological ER $Ca^{2+}$ loads, which is the opposite trend seen for the SERCA inhibitor phospholamban. DWORF also increased the ER $[Ca^{2+}]$ load (arrow). *$p<0.05$ versus CTRL.

## Relative affinity of DWORF binding to SERCA

To gain insight into the mechanism of SERCA activation by DWORF, we measured the relative binding of PLN and DWORF to SERCA in a cell environment. Fluorescent protein tags were fused to SERCA, PLN, and DWORF, and intermolecular fluorescence resonance energy transfer (FRET) was quantified in cells expressing pairs of fusion proteins. This method was recently used to show that PLN and DWORF had similar apparent affinities for SERCA in intact cells (*Makarewich et al., 2018*; *Singh et al., 2019*). In addition, we previously showed that PLN affinity for SERCA is sensitive to calcium, where the affinity was decreased by high calcium in permeabilized cells or during extended calcium elevations in rapidly paced cardiomyocytes (*Bidwell et al., 2011*). This apparent change in affinity suggests that PLN binds more avidly to the conformations of SERCA that prevail under resting conditions when cytosolic calcium is low. The enhanced interaction with calcium-free forms of SERCA offers a mechanistic explanation for the effect of PLN on the apparent calcium affinity of SERCA. PLN binds to SERCA in a groove formed by transmembrane segments M2, M6, and M9 (*Akin et al., 2013*; *Toyoshima et al., 2013*; *Winther et al., 2013*). Upon calcium binding to SERCA, M2 undergoes a conformational change that closes the inhibitory groove and forms the calcium-bound E1 state of SERCA. PLN appears to act as a competitive inhibitor of calcium binding by impeding groove closure and formation of the calcium-bound E1 state.

To determine how the SERCA-DWORF regulatory complex responds to changes in calcium, we permeabilized cells expressing Cer-SERCA and YFP-PLN or YFP-DWORF in solutions mimicking low (diastolic; 0.1 µM) or high (systolic; 3 µM) intracellular calcium concentrations (*Figure 5*). FRET was

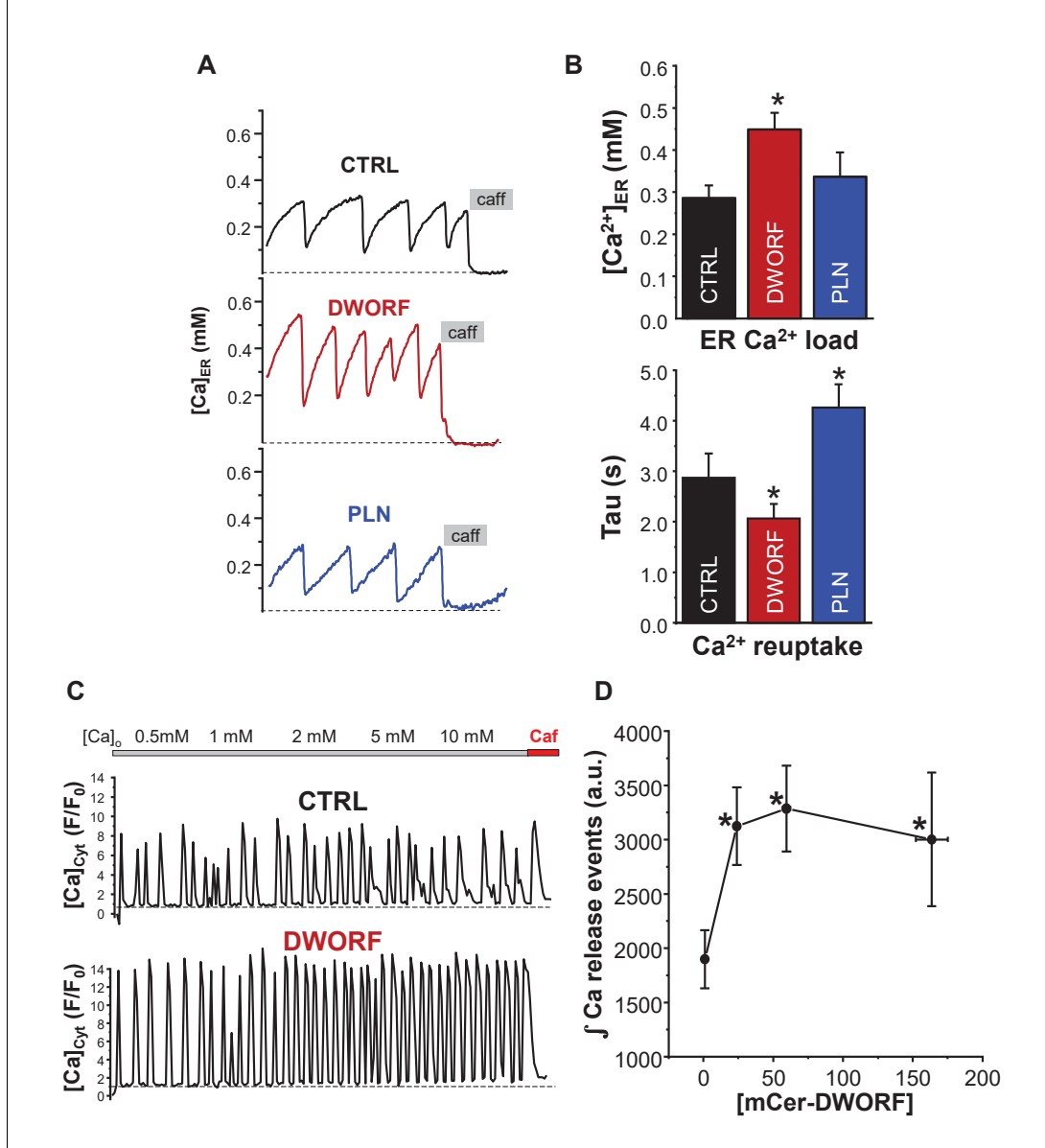

**Figure 4.** Effect of dwarf open reading frame (DWORF) on spontaneous calcium release from the endoplasmic reticulum (ER). (A, B) Co-expression of sarco-endoplasmic reticulum calcium pump (SERCA)2a and ryanodine receptor (RyR2) produced $Ca^{2+}$ waves due to spontaneous activation of RyR2 followed by SERCA $Ca^{2+}$ reuptake. DWORF increased the magnitude ($[Ca^{2+}]_{ER}$) and frequency (rate of recovery of $[Ca^{2+}]_{ER}$, Tau(s)) of spontaneous $Ca^{2+}$ waves, while phospholamban significantly decreased it. Similar effects of DWORF on spontaneous calcium release were observed when $Ca^{2+}$ waves were measured as cytosolic $Ca^{2+}$ fluctuations in intact cells. (C) Average amplitude and frequency of RyR-mediated $Ca^{2+}$ release events were significantly increased in DWORF-expressing cells. (D) The integral of RyR-mediated $Ca^{2+}$ release events was significantly increased in DWORF-expressing cells.

The online version of this article includes the following figure supplement(s) for figure 4:

**Figure supplement 1.** Examples of the experimental protocol used to measure sarco-endoplasmic reticulum calcium pump-mediated endoplasmic reticulum (ER) $Ca^{2+}$ uptake.

quantified for each cell and compared to that cell's level of expression of the YFP acceptor. As previously observed (*Kelly et al., 2008*), FRET between SERCA-PLN was lowest for cells expressing low levels of protein and increased to a maximal level of ~25% FRET efficiency for high-expressing cells (*Figure 5A*). The relationship was well-described by a hyperbolic fit that yielded the apparent dissociation constant ($K_D$) of the SERCA-PLN complex (*Figure 5C*). Permeabilization of cells in high

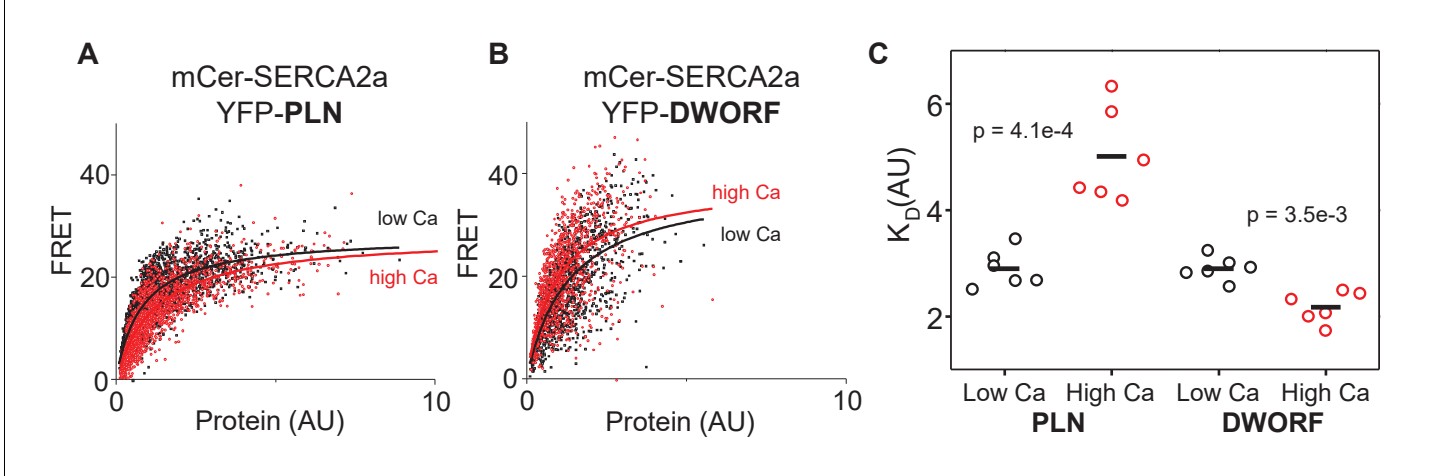

**Figure 5.** FRET analysis of sarco-endoplasmic reticulum calcium pump–dwarf open reading frame (SERCA-DWORF) interactions. The average acceptor sensitization FRET efficiency of cells co-transfected with mCer-SERCA2a and either (**A**) YFP-PLN or (**B**) YFP-DWORF. FRET efficiency was measured at high and low calcium concentrations to assess the relative affinity of phospholamban (PLN) and DWORF for the calcium-free and calcium-bound conformations of SERCA. (**C**) Hyperbolic fits to data provide quantification of the apparent dissociation constant ($K_D$) of the SERCA-PLN or SERCA-DWORF regulatory complexes. Ca decreases the apparent affinity of PLN for SERCA, yet it increases the affinity of DWORF for SERCA.

calcium yielded a binding curve that was right-shifted to higher protein concentrations, suggesting a decrease in SERCA-PLN binding affinity compared to low calcium conditions (*Bidwell et al., 2011*). In contrast, the SERCA-DWORF regulatory complex showed the opposite response to calcium, with a small left-shift of the hyperbolic FRET versus protein concentration curve (*Figure 5B*). In this case, the binding curve was left-shifted to lower protein concentrations, suggesting an increase in SERCA-DWORF binding affinity at high calcium (*Figure 5C*). The data indicate that high calcium decreases the affinity of PLN for SERCA, yet it has the opposite effect on DWORF where it increases the affinity for SERCA. These data provide a mechanistic framework for DWORF activation of SERCA observed in vitro (*Figure 2*) and in cells (*Figures 3* and *4*).

## Structure of DWORF

To gain molecular insight into the activation behavior described above, we generated two molecular models of DWORF, one as a continuous α-helix and another as an N-terminal α-helix (residues 1–13), a flexible linker (residues 14–16; centered around $Pro^{15}$), and a C-terminal α-helix (residues 17–35). The models were embedded in a lipid bilayer and equilibrated using MD simulations for 4 μs (*Figure 6*). Both starting models quickly converged on a bent helix or helix-linker-helix structure, though the overall structure was dynamic in the simulations and varied somewhat depending on the starting model. A similar behavior has been observed with SLN (*Aguayo-Ortiz et al., 2020*). The MD simulations support the helix-linker-helix model, with a bent or unwound region centered on $Pro^{15}$, as the more stable and probable structure of DWORF (*Figure 2B*). This model was maintained over 4 μs of MD simulations. To further rationalize this model, we carried out predictions of the secondary structure (*Buchan and Jones, 2019*; *Deléage, 2017*; *Drozdetskiy et al., 2015*; *Torrisi et al., 2019*) and the transmembrane region (*Buchan and Jones, 2019*; *Tusnády and Simon, 2001*; *Krogh et al., 2001*; *Rost and Liu, 2003*), as well as hydrophobic moment analysis of the DWORF peptide using only the amino acid sequence (*Figure 1*). The secondary structure predictions suggested that residues 9–35 are likely to be found as an α-helix, and prediction of the transmembrane domain suggested that residues 10–33 reside in the membrane.

To evaluate the N-terminal juxtamembrane helix in the helix-linker-helix model of DWORF and the potential 'helix breaker' property of $Pro^{15}$, we carried out hydrophobic moment analysis of the DWORF amino acid sequence (*Figures 1* and *7*). Hydrophobic moment analysis revealed an amphipathic helix comprising residues 1–13, which is remarkably like the N-terminus of PLN. However, DWORF has a shorter sequence following the proline residue ($Pro^{21}$ in PLN and $Pro^{15}$ in DWORF), which includes only the transmembrane domain. PLN has a polar helical region (residues $Gln^{22}$ to

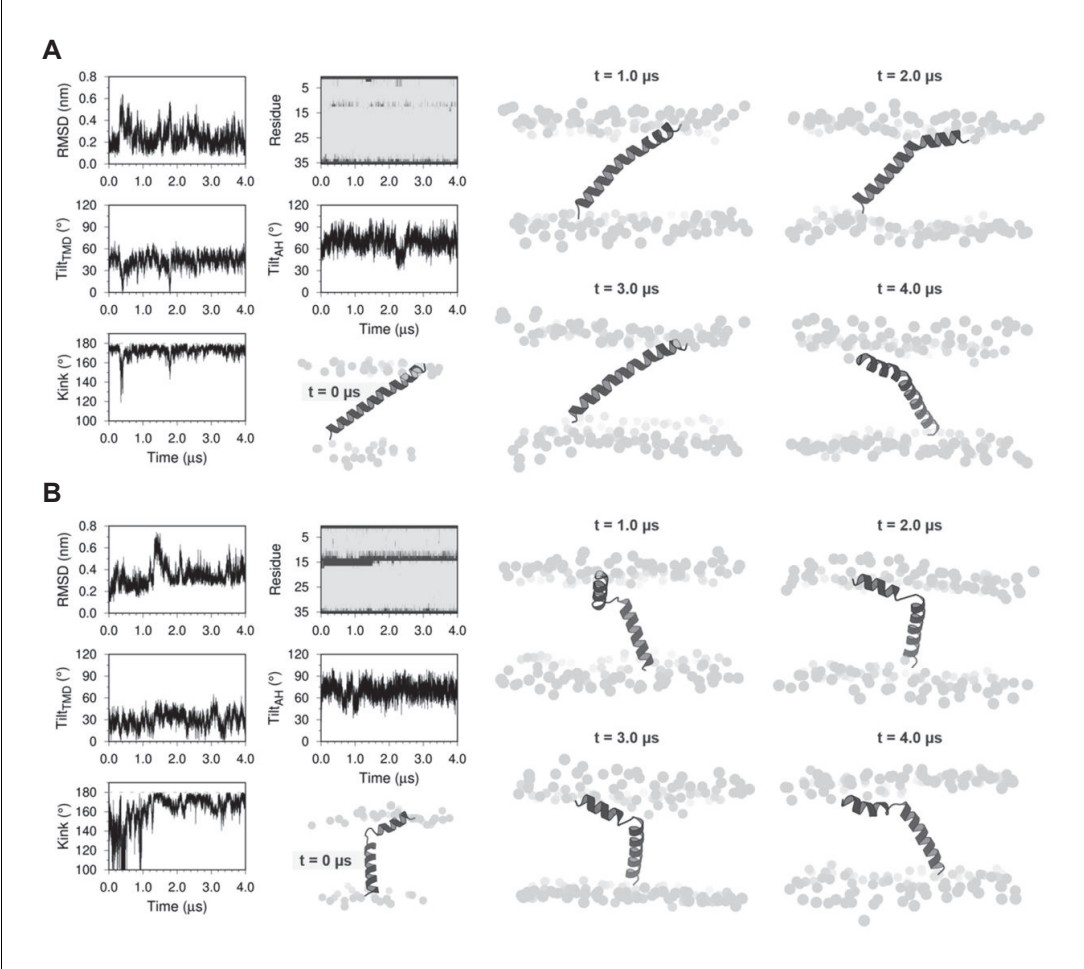

**Figure 6.** Molecular dynamics (MD) simulations of dwarf open reading frame (DWORF) modeled as a continuous helix (**A**) and as a helix-linker-helix (**B**). Shown are snapshots during the simulations (0, 1, 2, 3, and 4 µs), as well as the RMSD (nm), kink angle, per-residue secondary structure (coil in dark gray and helix in light gray), and tilt angles of the transmembrane helix domain (TMD) and the amphipathic helix (AH). Notice that both simulations support a helix-linker-helix structure of DWORF. The MD simulation in (**B**) indicates that the helix-linker-helix structure is maintained, though it is dynamic during the time course of the simulation.

Asn[30]) that follows the proline residue and precedes the transmembrane domain (residues 31–51), and this feature is absent in DWORF. The presence of the amphipathic helix allows hydrophobic residues to be located on one side of the helix and hydrophilic residues on the other. In the molecular model, the hydrophobic residues are oriented towards the lipid bilayer, while the hydrophilic residues are exposed to the aqueous solvent. As a result, DWORF modulates the local lipid bilayer (*Figure 7B*). The amphipathic juxtamembrane region is also observed in PLN (PDB code 2KYV; *Verardi et al., 2011*) and the amyloid precursor protein (PDB code 2LLM; *Nadezhdin et al., 2011*). Thus, the formation of this juxtamembrane helix suggests that DWORF may mimic some of the structural features of PLN.

## Molecular model of the SERCA-DWORF complex

To consider the implications of the helix-linker-helix model of DWORF, we generated structures of DWORF bound to the inhibitory groove of SERCA using protein-protein docking and MD simulations. DWORF was docked to the SERCA-PLN complex (PDB: 4KYT; *Akin et al., 2013*) with Pro[15] of DWORF aligned with Asn[34] of PLN (*Figure 8*). Three replicate SERCA-DWORF complexes were generated using the most populated structures of DWORF based on root-mean-square deviation (RMSD) clustering analysis of the MD simulations. During the 1 µs MD simulation times,

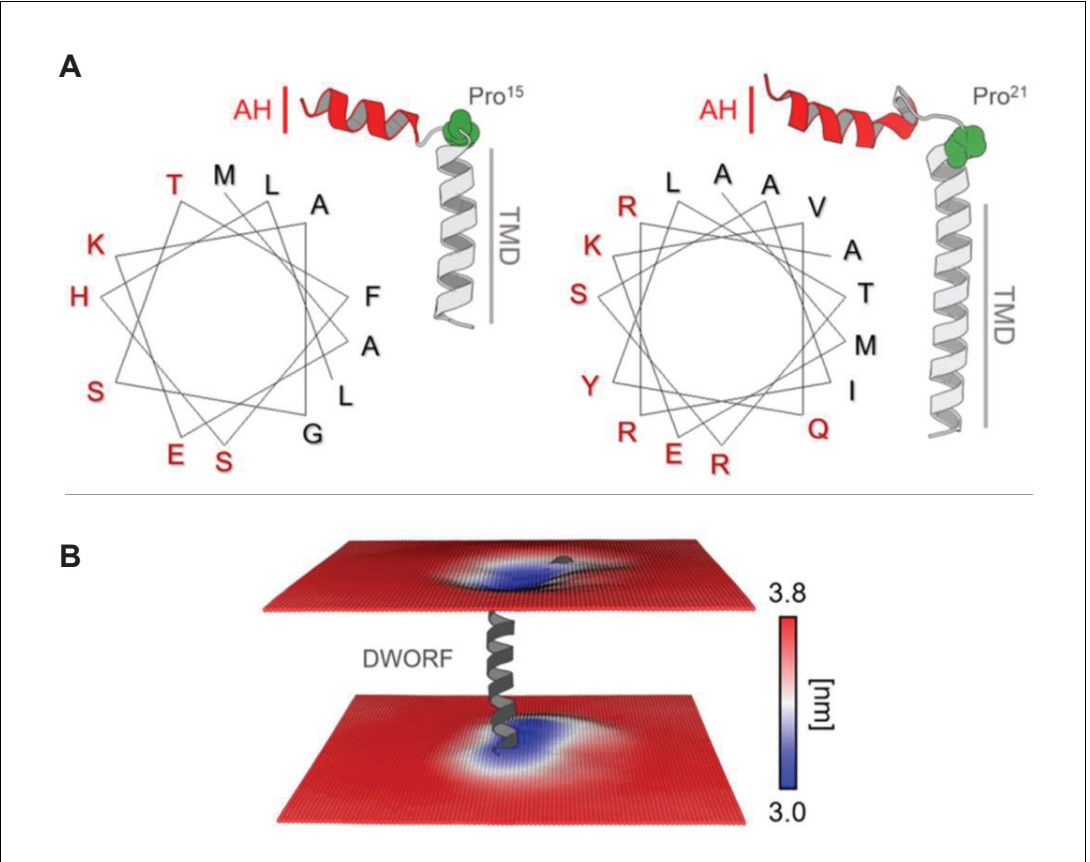

**Figure 7.** Helix-linker-helix model of dwarf open reading frame (DWORF). (**A**) Hydrophobic moment analysis revealed amphipathic helices at the N-terminus of DWORF (left panel) and phospholamban (PLN) (right panel). Notice the similar distribution of polar (red) and apolar (black) residues in the helix-kink structure of DWORF and PLN N-terminal to a proline residue (Pro[15] in DWORF and Pro[21] in PLN). (**B**) Time-averaged local membrane thickness in the molecular dynamics simulation of the DWORF helix-linker-helix model. Notice how the bilayer becomes thinner around the short transmembrane helix and amphipathic helix of DWORF. The maximum and minimum widths of the lipid bilayer are indicated by the blue-white-red color-coded bar.

the SERCA-DWORF complexes were maintained and the interaction remained stable (*Figure 8A*) with the exception of MD replicate MD2, which lost contact with DWORF for approximately 25% of the simulation time. However, the SERCA-DWORF interaction was reestablished. While the main focus of this study was not the precise molecular interactions of the complex, there are some interactions between DWORF and SERCA that appear to stabilize the complex (*Figure 8B*). Pro[15] and Ile[16] of DWORF interact with Trp[107] on M2 of SERCA. At the base of the transmembrane helix of DWORF, Tyr[31] packs against a short orthogonal helix at the base of M9 in the inhibitory groove of SERCA (Pro[952] and Ile[956]). In addition to these anchoring interactions, hydrophobic interactions dominate the interface, with the inhibitory groove of SERCA lined by DWORF hydrophobic residues including Leu[17], Ile[20], Ile[23], Val[24], Ile[27], Ile[28], and Ile[30]. The MD simulations also reveal that the bilayer thickness is modulated by DWORF in the vicinity of the inhibitory groove and the calcium access funnel of SERCA (*Figure 8C*).

The two main assumptions that underlie this model include (i) DWORF binds to the inhibitory groove of SERCA and (ii) Pro[15] of DWORF plays a role in disrupting interactions that contribute to SERCA inhibition (*Figure 9A,B*). First, DWORF binding to the inhibitory groove of SERCA is supported by the existing model that DWORF activates SERCA by displacing PLN (*Anderson et al., 2015*). Second, we tested the importance of Pro[15] in the activation of SERCA by measuring the ER calcium uptake rate and ER calcium load in the presence of a Pro[15]-Ala mutant of DWORF

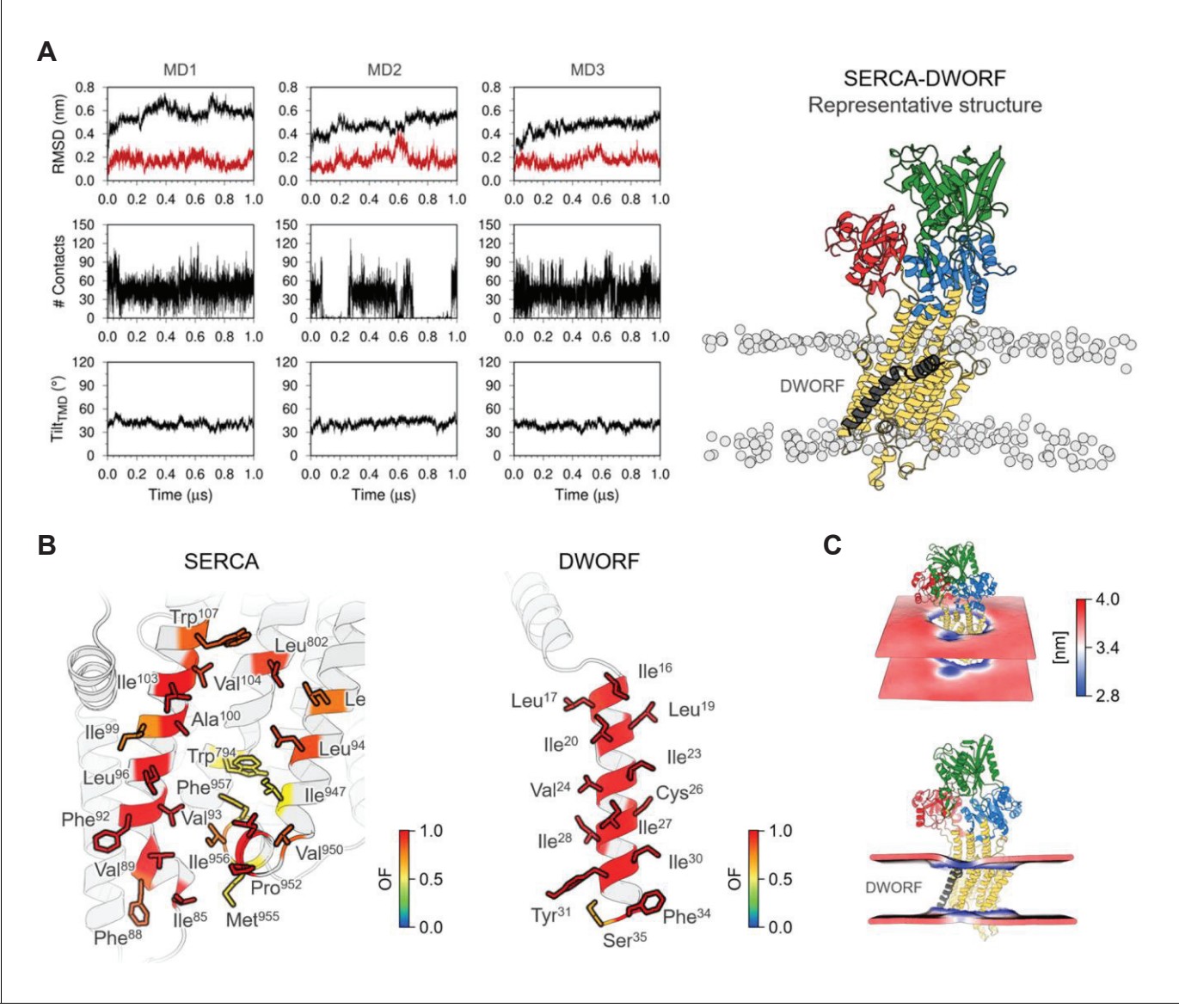

**Figure 8.** Molecular model for the interaction of sarco-endoplasmic reticulum calcium pump (SERCA) with dwarf open reading frame (DWORF). (**A**) Backbone RMSD, number of contacts between SERCA and DWORF, and tilt angle of DWORF transmembrane helix domain in the three 1 μs replicates of SERCA-DWORF molecular dynamics (MD) simulations. The figure to the right shows the representative structure of the SERCA-DWORF complex from the RMSD clustering analysis of the three simulations. SERCA is colored yellow, with the nucleotide-binding domain in green, the phosphorylation domain in blue, and the actuator domain in red. DWORF is shown in gray. (**B**) SERCA residues located within 3.0 Å of DWORF (left panel) and DWORF residues located within 3.0 Å of SERCA (right panel) colored by the computed occupancy fraction (OF) of the three independent simulations. Only the residues that presented an OF ≥ 0.6 are shown in the figure. (**C**) Representation of the averaged local membrane thickness of SERCA-DWORF complex calculated with the first replicate MD1.

(*Figure 9C*). The enhanced cellular calcium dynamics observed in the presence wild-type DWORF were abolished in the presence of the Pro[15]-Ala missense variant of DWORF.

## Discussion

### Functional effect of DWORF

The data presented here demonstrate that DWORF is a direct activator of SERCA, enhancing calcium-dependent ATPase activity of SERCA in reconstituted membranes and SERCA-dependent

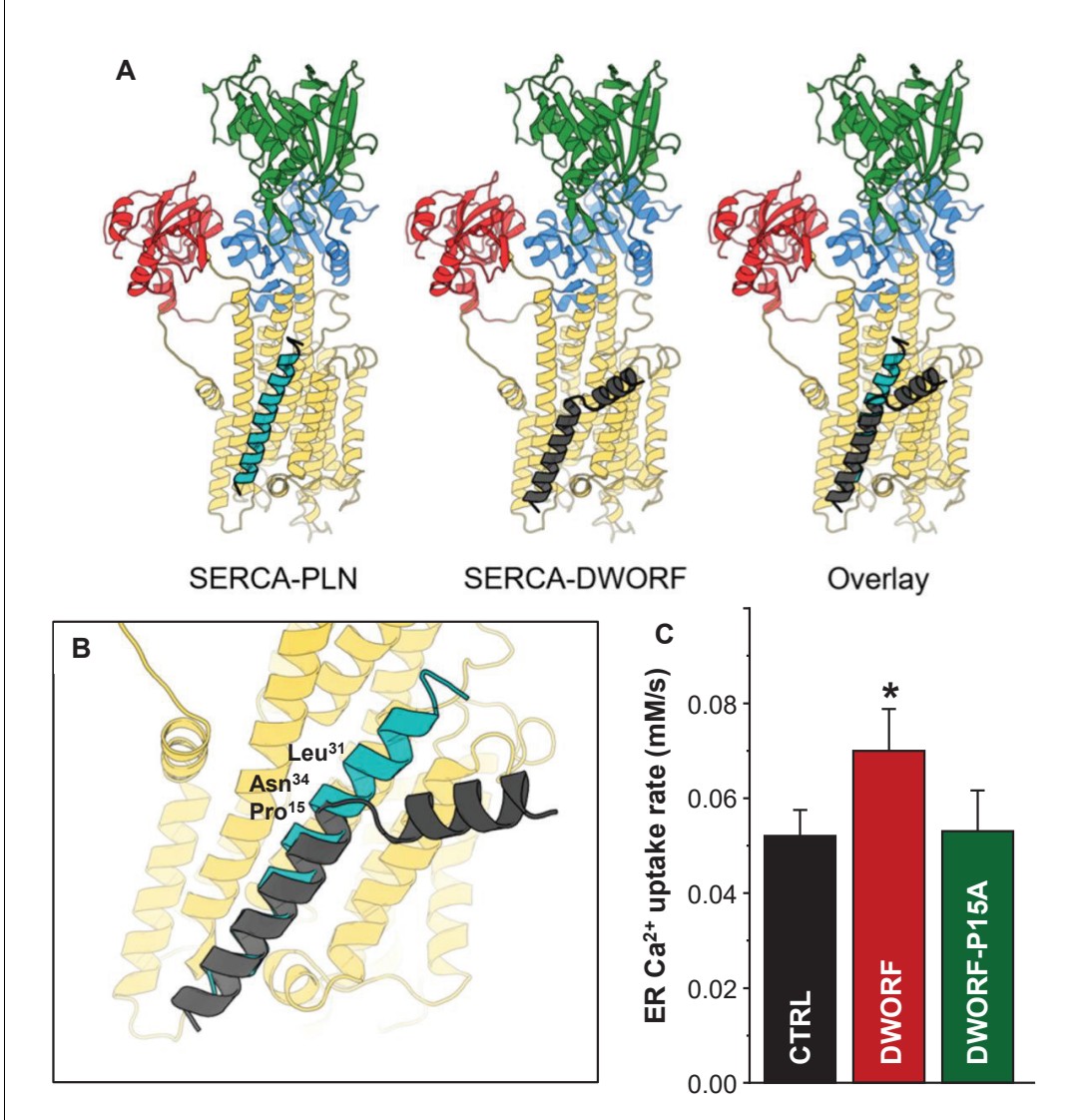

**Figure 9.** Molecular model for the interaction of sarco-endoplasmic reticulum calcium pump (SERCA) with dwarf open reading frame (DWORF). (**A**) SERCA-PLN, SERCA-DWORF, and the overlay are shown in cartoon format. The molecular model of DWORF (**Figure 6A**) was superimposed on the structure of the SERCA-PLN complex (PDB ID: 4KYT) according to the sequence alignment in **Figure 1**. SERCA is colored yellow, with the nucleotide-binding domain in green, the phosphorylation domain in blue, and the actuator domain in red. Phospholamban (PLN) is shown in cyan and DWORF in gray. (**B**) Close-up view of the SERCA-PLN and SERCA-DWORF complexes. The relative positions of Leu[31] and Asn[34] are indicated, as well as Pro[15] of DWORF, which aligns with Asn[34] of PLN. (**C**) Endoplasmic reticulum calcium uptake rate for the Pro15-Ala (P15A) mutant of DWORF compared to SERCA alone (CTRL) and SERCA in the presence of wild-type DWORF (DWORF). *p<0.01.

calcium dynamics in living cells. While DWORF displaces PLN from SERCA as previously reported (*Nelson et al., 2016*; *Makarewich et al., 2018*), we have shown that DWORF also has a direct effect on SERCA function that is opposite to the inhibitory properties of PLN. The opposing functions of DWORF and PLN represent a previously unknown regulatory axis for fine-tuning SERCA-dependent calcium homeostasis in response to demand for cardiac output. Previous studies of DWORF overexpression in mouse models with normal PLN expression level demonstrated that peak systolic calcium transient amplitude and SR calcium load were significantly increased with DWORF overexpression (*Nelson et al., 2016*; *Makarewich et al., 2018*). While these findings were attributed to DWORF displacement of PLN and relief of SERCA inhibition, these findings are also consistent with a dual mechanism that includes direct activation of SERCA by DWORF.

The membrane reconstitution system allowed us to assess the effect of DWORF on SERCA function in a controlled membrane environment. This SERCA-DWORF two-component system is well-suited for measurements of ATPase activity (*Ceholski et al., 2012*; *Trieber et al., 2009*; *Trieber et al., 2005*; *Gorski et al., 2013*; *Douglas et al., 2005*) and calcium transport (*Smeazzetto et al., 2017*) in the presence of regulatory subunits such as PLN and DWORF. PLN is a known inhibitor of SERCA in that it reduces the apparent affinity for calcium (*Figure 2A*). Herein, we measured SERCA ATPase activity in the presence of DWORF, which allowed assessment of both the apparent calcium affinity and maximal activity of SERCA. Previous studies of DWORF function focused on the apparent calcium affinity of SERCA (*Nelson et al., 2016*; *Makarewich et al., 2018*). At a near equimolar ratio, DWORF acted as a direct activator by increasing the maximal activity of SERCA 1.7-fold without an effect on the apparent calcium affinity (*Figure 2B*). The magnitude of the $V_{max}$ increase seen in the presence of DWORF (~1.7 fold) is similar to that seen in the presence of the small-molecule activator of SERCA, CDN1163 (~1.5-fold; *Kang et al., 2016*). Thus, the important feature of DWORF and PLN regulation in cardiac muscle is that the regulatory subunits have different functions. The primary effect of PLN is to alter the apparent calcium affinity of SERCA, and the primary effect of DWORF is to alter the maximal activity of SERCA. These distinct but complimentary functions allow for a fine level of control of SERCA-mediated calcium homeostasis.

## Cellular effect of DWORF

Membrane co-reconstitution systems can provide mechanistic insight into SERCA regulation; however, it is both an advantage and disadvantage that they lack the complexity of a cellular environment. HEK293 cells have been established as a novel cardiomimetic system for evaluating SERCA-dependent calcium-handling and its regulation by PLN (*Bovo et al., 2019*). This approach provides the complex environment of a cell, while avoiding the confounding effects of imaging cells during contractions. Co-expression of SERCA2a and RyR2 in this cell system causes periodic calcium transients (waves) similar to those that elicit contractions in cardiomyocytes. The HEK cell model lacks endogenous SERCA regulators, and it can be stably transfected with the exogenous regulators PLN and DWORF. This enabled the observation of ER calcium load, calcium uptake rate, and changes in the magnitude and frequency of calcium waves. These are SERCA-dependent cellular responses to changes in calcium concentration and the presence of SERCA regulatory subunits. This system was used to test the effects of DWORF on SERCA-dependent cellular calcium dynamics. As was observed for PLN, DWORF co-localized with SERCA in ER membranes (*Figure 3A*). DWORF increased two key parameters of SERCA function, ER calcium content (*Figure 3B*) and ER calcium uptake rate (*Figure 3C*). These parameters reflect the kinetics and thermodynamics of SERCA in the absence and presence of DWORF, and the effects were opposite to what is seen for PLN.

An alternative measure of ER calcium load and SERCA activity was provided by the cardiomimetic HEK system, which generates periodic calcium waves due to spontaneous calcium release by RyR2 followed by SERCA calcium reuptake. Calcium release via RyR2 is known to depend on ER calcium load, which is determined by SERCA activity. Thus, the cytosolic calcium concentration during periodic calcium waves is an indirect measure of SERCA function in the absence and presence of DWORF (*Figure 4A, B*). DWORF increased the amplitude and frequency of RyR-mediated calcium release events (*Figure 4C, D*). The SERCA-mediated calcium reuptake rate, ER calcium load, and calcium waves were all increased in DWORF-expressing cells, strongly suggesting that DWORF directly activates SERCA and increases its catalytic efficiency. These cellular data (*Figures 3* and *4*) agree with the activity data from reconstituted membranes containing SERCA and DWORF (*Figure 2*).

This raised the question, what is the mechanism by which DWORF activates SERCA? To address this, FRET efficiency between SERCA-PLN and SERCA-DWORF was measured at high and low calcium concentrations to assess the relative affinity of PLN and DWORF for the calcium-bound and calcium-free conformations of SERCA (*Figure 5A, B*). The data provided relative measures of the dissociation constants ($K_D$) of the SERCA-PLN and SERCA-DWORF complexes (*Figure 5C*). Previous studies have shown that PLN remains associated with SERCA even in the presence of calcium (e.g., *Bidwell et al., 2011*; *Dong and Thomas, 2014*; *Mueller et al., 2004*; *Li et al., 2004*). Consistent with these observations, elevated calcium did not abolish binding of PLN to SERCA, instead it reduced the apparent affinity of PLN for SERCA (*Figure 5A, C*). In contrast, calcium increased the apparent affinity of DWORF for SERCA (*Figure 5B, C*), revealing that DWORF prefers to interact with conformations of SERCA that prevail at high calcium. In the context of the current model of

DWORF as a competitive inhibitor of PLN binding to SERCA, the data suggest that DWORF would compete more effectively when calcium is elevated, helping to relieve SERCA inhibition each time that increased transport function is required. In addition, the inverted calcium dependence of the SERCA-DWORF interaction offers an explanation for the direct activation of SERCA by DWORF observed in reconstituted membranes (*Figure 2*) and cellular calcium dynamics (*Figures 3* and *4*). While PLN inhibition involves preferential interaction and stabilization of SERCA enzymatic states that prevail at low calcium (e.g., *Akin et al., 2013*; *Asahi et al., 2000*; *Jones et al., 2002*), DWORF preferentially binds and stabilizes SERCA states that are populated during calcium elevations. This DWORF interaction enhances the kinetics of rate-limiting steps in the SERCA transport cycle.

### Unique structure of DWORF

A critical step was to evaluate the structure of DWORF in comparison to the well-characterized SERCA regulatory subunits PLN (*Akin et al., 2013*; *Verardi et al., 2011*) and SLN (*Toyoshima et al., 2013*; *Winther et al., 2013*; *Mascioni et al., 2002*). Both SLN and PLN form continuous transmembrane helices with well-defined orientations in the membrane. The cytoplasmic domain of PLN is longer than that found in SLN and DWORF, and it lies along the membrane surface in the structure determined by NMR spectroscopy (PDB ID: 2KYV; *Verardi et al., 2011*). In the X-ray crystal structures of the SERCA-PLN (*Akin et al., 2013*) and SERCA-SLN (*Toyoshima et al., 2013*; *Winther et al., 2013*) complexes, PLN and SLN are found as continuous transmembrane helices (residues ~ 24–48 and ~1–31, respectively), though the cytoplasmic domain of PLN was not resolved. The helical transmembrane domains of PLN and SLN facilitate the structural interactions in the SERCA-bound states, and it is a critical feature of these peptides. In contrast, the DWORF transmembrane domain appears to be discontinuous (residues Leu[17] to Ser[35]), with a break at Pro[15] and an N-terminal helix that lies along the membrane surface (residues Met[1] to Leu[13]).

The molecular structure of the SERCA-DWORF complex remains unknown. The current hypothesis is that DWORF binds to the inhibitory groove and displaces PLN (*Nelson et al., 2016*), and that DWORF and PLN have similar affinities for SERCA (*Singh et al., 2019*). That said, the proposed helix-linker-helix structure of DWORF (*Figure 6s* and *7*) allows us to speculate about potential mechanisms. The break at Pro[15] of DWORF occurs at a critical location for PLN and SLN inhibition of SERCA (*Figures 1* and *9*). Asn[34] of PLN (Asn[11] in SLN) is an essential residue for SERCA inhibition, and it is positioned to interact with SERCA by the continuous transmembrane helix of PLN. Thus, if DWORF binds to SERCA and replaces PLN, DWORF lacks Asn[34] and N-terminal residues of PLN that contribute to SERCA inhibition (*Figure 9*). This model offers an explanation for why DWORF itself does not inhibit SERCA – the discontinuous transmembrane helix and the substitution of a proline residue are inconsistent with structural features known to contribute to SERCA inhibition. The model also offers an explanation for how DWORF activates SERCA. We have previously proposed that modulation of the lipid bilayer by PLN is a mechanism for enhancing SERCA maximal activity (*Glaves et al., 2019*). DWORF binding to the inhibitory groove of SERCA would be expected to modulate the lipid bilayer as suggested by the MD simulations (*Figure 8C*). Importantly, the modulation of the lipid bilayer observed in the MD simulations is more pronounced on the cytoplasmic side of the membrane where it could impact the dynamics of transmembrane helices (e.g., M1 and M2) that form the calcium access funnel. The displacement of PLN, the absence of key inhibitory interactions (two essential residues of PLN include Leu[31] and Asn[34]; *Trieber et al., 2009*; *Kimura et al., 1997*), and perturbation of the membrane bilayer provide rationales for the higher maximal activity of SERCA in the presence of DWORF.

## Materials and methods

**Key resources table**

| Reagent type (species) or resource | Designation | Source or reference | Identifiers | Additional information |
|---|---|---|---|---|
| Strain, strain background (*Escherichia coli*) | BL21(DE3) | Sigma-Aldrich | CMC0016 | Chemically competent cells |

*Continued on next page*

*Continued*

| Reagent type (species) or resource | Designation | Source or reference | Identifiers | Additional information |
|---|---|---|---|---|
| Recombinant DNA reagent | pMAL c2x DWORF plasmid | This study | | MBP-DWORF fusion; H.S. Young lab |
| Peptide, recombinant protein | Human DWORF | This study | NCBI: NP_001339058.1 | Purified human DWORF peptide; H.S. Young lab |
| Chemical compound, drug | CDN1163; 4-(1-methylethoxy)-N-(2-methyl-8-quinolinyl)-benzamide | Sigma-Aldrich | SML1682 | Previously published SERCA activator (PMID:26702054) |
| Chemical compound, drug | $C_{12}E_8$; octaethylene glycol monododecyl ether | Nikkol | BL-8SY | Detergent for solubilizing SERCA |
| Chemical compound, drug | Egg PC; L-α-phosphatidylcholine (Egg, Chicken) | Avanti | 840051 | Natural lipid |
| Chemical compound, drug | Egg PE; L-α-phosphatidylethanolamine (Egg, Chicken) | Avanti | 840021 | Natural lipid |
| Chemical compound, drug | Egg PA; L-α-phosphatidic acid (Egg, Chicken) | Avanti | 840101 | Natural lipid |
| Chemical compound, drug | Ionomycin, calcium salt | Sigma-Aldrich | I3909 | Facilitates the transport of calcium across the plasma membrane |
| Chemical compound, drug | Ruthenium red | Sigma-Aldrich | R2751 | RyR inhibitor |
| Chemical compound, drug | Caffeine | Sigma-Aldrich | C0750 | RyR2 agonist |
| Recombinant DNA reagent | pEGFP_RyR2 | Donated by Dr. C. George (PMID:15047862) | | Mammalian expression construct containing GFP fused RyR2 |
| Cell line (HEK 293) | T-Rex-293 Cell Line | Thermo Fisher Scientific | R71007 | Mammalian cell line for the stable expression of proteins of interest |
| Recombinant DNA reagent | EYFP-PLN | *Singh et al., 2019* | | Constitutively fluorescent green/yellow fluorescent protein fusion construct |
| Recombinant DNA reagent | EYFP-DWORF | *Singh et al., 2019* | | Constitutively fluorescent green/yellow fluorescent protein fusion construct |
| Recombinant DNA reagent | mCerulean-SERCA | *Bovo et al., 2019* | | Fluorescent protein fusion construct |
| Cell line (HEK 293) | AAV-293 cells | Agilent Technologies | 240073 | HEK293 cell line optimized for AAV transfection |

## Materials

All reagents were of the highest purity available: octaethylene glycol monododecyl ether ($C_{12}E_8$; Nikkol Group, Nikko Chemicals Co., Ltd., Tokyo, Japan); egg yolk phosphatidylcholine (EYPC), phosphatidylethanolamine (EYPE), and phosphatidic acid (EYPA) (Avanti Polar Lipids, Alabaster, AL); all reagents used in the coupled enzyme assay include NADH, ATP, PEP, lactate dehydrogenase, and pyruvate kinase (Sigma-Aldrich, Oakville, ON Canada).

## Co-reconstitution of DWORF and SERCA

Recombinant human DWORF was expressed as a maltose-binding protein (MBP) fusion with a TEV cleavage site for removal of MBP. DWORF was purified by a combination of organic extraction (chloroform-isopropanol-water) and reverse-phase HPLC (*Douglas et al., 2005*). Purified DWORF was

stored as lyophilized thin films (63 µg aliquots). SERCA1a was purified from rabbit skeletal muscle SR. For co-reconstitution, lyophilized DWORF (63 µg) was suspended in a 100 µl mixture of trifluoroethanol-water (5:1) and mixed with lipids (360 µg EYPC and 40 µg EYPA) from stock chloroform solutions. The peptide-lipid mixture was dried to a thin film under nitrogen gas and desiccated under vacuum overnight. The peptide-lipid mixture was hydrated in buffer (20 mM imidazole pH 7.0; 100 mM NaCl; 0.02% NaN$_3$) at 50°C for 15 min, cooled to room temperature, and detergent-solubilized by the addition of C$_{12}$E$_8$ (0.2% final concentration) with vigorous vortexing. Detergent-solubilized SERCA1a was added (300 µg in a total volume of 300 µl) and the reconstitution was stirred gently at room temperature. Detergent was slowly removed by the addition of SM-2 Bio-Beads (Bio-Rad, Hercules, CA) over a 4 hr time course (final weight ratio of 25 Bio-Beads to one detergent). Following detergent removal, the reconstitution was centrifuged over a sucrose step-gradient (20–50% layers) for 1 hr at 100,000 g. The reconstituted proteoliposomes at the gradient interface were removed, flash-frozen in liquid-nitrogen, and stored at −80°C. The final molar ratio was 120 lipids to 2 DWORF to 1 SERCA. To ensure the proper ratio of ~2 DWORF per SERCA in the reconstituted proteoliposomes, quantitative SDS-PAGE was used with known quantities of SERCA and DWORF proteins as standards (*Trieber et al., 2009*; *Trieber et al., 2005*; *Young et al., 2001*).

## ATPase activity assays of SERCA reconstitutions

ATPase activity of the co-reconstituted proteoliposomes was measured by a coupled-enzyme assay over a range of calcium concentrations from 0.1 µM to 10 µM (*Trieber et al., 2009*; *Trieber et al., 2005*). The assay has been adapted to a 96-well format utilizing Synergy 4 (BioTek Instruments) or SpectraMax M3 (Molecular Devices) microplate readers. Data points were collected at 340 nm wavelength, with a well volume of 155 µl containing 10–20 nM SERCA at 30°C (data points collected every 28–39 s for 1 hr). The reactions were initiated by the addition of proteoliposomes to the assay solution. The V$_{max}$ (maximal activity) and K$_{Ca}$ (apparent calcium affinity) were determined based on nonlinear least-squares fitting of the activity data to the Hill equation (Sigma Plot software, SPSS Inc, Chicago, IL). Errors were calculated as the standard error of the mean for a minimum of four independent reconstitutions.

## In-cell calcium uptake and spontaneous calcium release methods

pCMV R-CEPIA1er was a gift from Dr. Masamitsu Iino (Addgene, USA). The vector encoding the human RyR2 cDNA fused to GFP at the N-terminus was kindly provided by Dr. Christopher George (University of Cardiff, UK). The vector encoding human SERCA2a cDNA was kindly provided by Dr. David Thomas (University of Minnesota, USA). The SERCA2a cDNA was cloned into the mCerluean-M1 modified plasmid (Addgene, USA), yielding SERCA2a fused to a modified Cerulean fluorescent protein (mCer) at the N-terminus. SERCA2a cDNA was also cloned into the inducible expression vector pcDNA5/FRT/TO for the generation of SERCA2a stable cell line. The sequences were all verified by single pass primer extension analysis (ACGT Inc, USA).

## Generation of SERCA2a stable cell line

Stable inducible Flp-In T-Rex-293 cell line expressing SERCA2a was generated using the Flp-In T-REx Core Kit (Invitrogen, USA) as described (*Bovo et al., 2019*). Flp-In T-REx-293 cells were co-transfected with the pOG44 vector encoding the Flp recombinase and the expression vector pcDNA5/FRT/TO containing the SERCA2a cDNA. 48 hr after transfection, the growth medium was replaced with a selection medium containing 100 µg/ml hygromycin. The hygromycin-resistant cell foci were selected and expanded. Stable cell lines were cultured in high glucose Dulbecco's modified Eagle's medium (DMEM) supplemented with 100 units/ml penicillin, 100 mg/ml streptomycin, and 10% fetal bovine serum (FBS) at 5% CO$_2$ and 37°C. Expression of SERCA2a in the stable cell line was verified by western blot analysis 48 hr after induction of recombinant pump expression with 1 µg/ml tetracycline. This strategy results in SERCA2a expressed 10-fold over endogenous SERCA2b (*Bovo et al., 2019*), so calcium uptake is dominated by the exogenous cardiac isoform. MCer-DWORF expression in these cells showed a similar pattern as CEPIA-1er (*Figure 4A*), indicating that DWORF is preferentially localized in the ER membrane. Since CICR depends on RyR2 expression level, we analyzed calcium waves only in cells that had similar GFP-RyR2 signal.

## Confocal microscopy

Changes in the luminal ER $[Ca^{2+}]$ ($[Ca^{2+}]_{ER}$) and cytosolic $[Ca^{2+}]$ ($[Ca^{2+}]_{Cyt}$) were measured with laser scanning confocal microscopy (Radiance 2000 MP, Bio-Rad, UK) equipped with a $\times 40$ oil-immersion objective lens (N.A. = 1.3). T-Rex-293 cells expressing SERCA2a were transiently co-transfected with plasmids containing the cDNA of RyR2, DWORF, and R-CEPIA1er. Experiments were conducted 48 hr after transfection to obtain the optimal level of recombinant protein expression. The surface membrane was permeabilized with saponin (0.005%). Experiments were conducted after wash out of saponin with a solution of 100 mM K-aspartate, 15 mM KCl, 5 mM $KH_2PO_4$, 5 mM MgATP, 0.35 mM EGTA, 0.22 mM $CaCl_2$, 0.75 mM $MgCl_2$, 10 mM HEPES (pH 7.2), and 2% dextran (MW: 40,000). Free $[Ca^{2+}]$ and $[Mg^{2+}]$ of this solution were 200 nM and 1 mM, respectively.

## $[Ca^{2+}]_{ER}$ measurements

$[Ca^{2+}]_{ER}$ was recorded as changes in fluorescence intensity of the genetically encoded ER-targeted $Ca^{2+}$ sensor R-CEPIA1er (*Bovo et al., 2016*). R-CEPIA1er was excited with a 514 nm line of the argon laser and signal was collected at >560 nm. Line-scan images were collected at a speed of 10 ms/line. The R-CEPIA1er signal (F) was converted to $[Ca^{2+}]_{ER}$ by the following formula: $[Ca^{2+}]_{SE} = K_d \times [(F - F_{min})/(F_{max} - F)]$. $F_{max}$ was recorded in 5 mM $Ca^{2+}$ and 5 μM ionomycin and $F_{min}$ was recorded after ER $Ca^{2+}$ depletion with 5 mM caffeine. The $K_d$ ($Ca^{2+}$ dissociation constant) was 564 μM (*Mekahli et al., 2011*). SERCA-mediated $Ca^{2+}$ uptake was calculated as the first derivative of $[Ca^{2+}]_{ER}$ refilling ($d[Ca^{2+}]_{ER}/dt$) after RyR2 inhibition with ruthenium red (15 μM) and tetracaine (1 mM). RyR2-independent $Ca^{2+}$ leak was analyzed as the first derivative of $[Ca^{2+}]_{ER}$ decline ($d[Ca^{2+}]_{ER}/dt$) after simultaneous inhibition of RyR2 and SERCA. ER $Ca^{2+}$ uptake and $Ca^{2+}$ leak rates were plotted as a function of $[Ca^{2+}]_{ER}$ to analyze maximum ER $Ca^{2+}$ uptake rate and maximum ER $Ca^{2+}$ load. All 2-D images and line-scan measurements for $[Ca^{2+}]_{ER}$ were analyzed with ImageJ software (NIH, USA).

## Cytosolic $[Ca^{2+}]$ ($[Ca^{2+}]_i$) measurements

$[Ca^{2+}]_i$ was measured in intact cells with the high-affinity $Ca^{2+}$ indicator Fluo-4 (Molecular Probes/Invitrogen, Carlsbad, CA). To load the cytosol with Fluo-4, cells were incubated at room temperature with 10 μM Fluo-4 AM for 15 min in Tyrode solution (140 mM NaCl, 4 mM KCl, 0.5 mM $CaCl_2$, 1 mM $MgCl_2$, 10 mM glucose, 10 mM HEPES, pH 7.4), followed by a 20 min wash. Fluo-4 was excited with the 488 nm line of an argon laser and the emission signal collected at wavelengths above 515 nm. Spontaneous $Ca^{2+}$ waves were measured at different $[Ca^{2+}]$ (0.5, 1, 2, 5, and 10 mM). In the end of each experiment, caffeine was applied to induce maximal ER $Ca^{2+}$ release.

## Statistics

Data are presented as mean ± standard error of the mean (SEM) of n measurements. Statistical comparisons between groups were performed with the Student's t test for unpaired data sets. Differences were considered statistically significant at $p < 0.05$. Statistical analysis and graphical representation of averaged data were carried out on OriginPro7.5 software (OriginLab, USA).

## In-cell FRET methods

### Fluorescence resonance energy transfer measurements

Acceptor sensitization FRET was quantified as previously described (*Bidwell et al., 2011*). AAV 293 cells were cultured in DMEM cell culture medium supplemented with 10% FBS (ThermoScientific, Waltham, MA). Following culture, cells were transiently transfected using either MBS mammalian transfection kit (Agilent Technologies, Stratagene, La Jolla, CA) or Lipofectamine 3000 transfection kit (Invitrogen) with either EYFP-PLN or DWORF, and mCerulean (mCer)-SERCA constructs in a 1:5 molar plasmid ratio with the fluorescent protein fused via a five amino acid linker to the N-terminus (*Makarewich et al., 2018*; *Singh et al., 2019*; *Autry et al., 2011*; *Robia et al., 2007*). Cells were then plated 24 hr before imaging in four-well chambered coverglass plates coated with poly-D-lysine and imaged utilizing wide-field fluorescent microscopy. Cells were imaged on a Nikon Eclipse Ti 2 equipped with a Photometrix Prime 95B CMOS camera (Tucson, AZ) and Lumencor Spectra X (Beaverton, OR). Imaging was done in a permeabilization buffer containing 120 mM potassium aspartate, 15 mM potassium chloride, 5 mM magnesium ATP, 0.75 mM magnesium chloride, 2% dextran, 5

mM potassium phosphate, 2 mM EGTA, 20 mM HEPES, and either 0 or 1.7 mM added calcium chloride. Cells were permeabilized in 0.005% saponin buffer for 1 min and then washed twice with buffer before imaging. Data were collected using a 20× 0.75 N.A. objective using 50 ms exposure times for all channels and analyzed using in FIJI using a macro selecting cells with a CFP intensity above 150 AU above background, circularity of 0.4–1 AU, and size of 500–2500 pixels with a rolling background size of 200. FRET efficiency was calculated according to $E_{app} = I_{DA} – a(I_{AA}) – d(I_{DD}) /(I_{DA} – a(I_{AA}) + (G-d)) I_{DD}$ (*Zal and Gascoigne, 2004*), where $I_{DA}$ is the intensity of fluorescence of acceptor emission with donor excitation; $I_{AA}$ is the intensity of acceptor fluorescence with acceptor excitation; and $I_{DD}$ is the fluorescence intensity of donor emission with donor excitation. G represents the ratio of sensitized emission to the corresponding amount of donor recover during acceptor photobleaching and acts as a correction factor, which is constant for a given fluorophore and image conditions. Constants a and d are bleed-through constants calculated from $a = I_{DA}/ I_{AA}$ for a control sample transfected with only YFP-labeled SERCA, and $d = I_{DA}/ I_{DD}$ for a sample transfected with only Cer-labeled SERCA. These values were determined to be a = 0.185, d = 0.405, and G = 2.782. FRET intensity of each cell was then compared to the cell's YFP intensity, which was used as a measure of protein expression ([PLN]). This leads to estimates of apparent dissociation constant ($K_D$), YFP intensity at ½ $FRET_{max}$, and intrinsic FRET of the SERCA-PLN complex, which is equal to the $FRET_{max}$. Data were fit with a hyperbola of the function $FRET = (FRET_{max})([PLN])/(K_D + [PLN])$.

## Cell lines

The Flip-In T-Rex Cell Line was purchased from Thermo Fisher Scientific. The Flip-In T-Rex Cell Line is derived from 293 human embryonic kidney cells. The 293 parental cell line was obtained from the American Type Culture Collection, and every batch of cells has a Certificate of Analysis. Low passage AAV-293 cells (a type of HEK-293 cell) were obtained from Agilent Technologies. All cells were routinely screened for *Mycoplasma* using DAPI staining and PCR.

## Molecular modeling of DWORF

The protein structure homology-modeling program MODELLER (*Webb and Sali, 2016*) was used to generate molecular models of DWORF as a continuous α-helix (residues 1–35) and a helix-linker-helix (residues 1–13 modeled as an α-helix; residues 14–16 as random coil; and residues 17–35 as an α-helix). In evaluating the amino acid sequence of DWORF, we carried out secondary structure prediction using PSIPRED (*Buchan and Jones, 2019*), MLRC (*Deléage, 2017*), JPRED v4.0 (*Drozdetskiy et al., 2015*), and Porter v5.0 (34). We performed transmembrane region predictions using MEMSAT-SVM (*Buchan and Jones, 2019*), HMMTOP (*Tusnády and Simon, 2001*), TMHMM v2.0 (*Krogh et al., 2001*), and PredictProtein (*Rost and Liu, 2003*). Finally, we carried out hydrophobic moment analysis to identify regions of amphipathic helices using PEPWHEEL (*Di Scala et al., 2014*) and HMOMENT (*Rice et al., 2000*). The SERCA-DWORF complex was modeled based on the SERCA-PLN structure (PDB: 4KYT; *Akin et al., 2013*) using PyMol v1.7 (*Schrodinger, LLC, 2015*) and the SwissModel server (*Waterhouse et al., 2018*).

## Molecular dynamics simulations

The DWORF and SERCA-DWORF models were inserted in a 1-palmitoyl-2-oleoyl-sn-glycero-3-phosphocholine (POPC) lipid bilayer containing a total 180 and 370 lipid molecules, respectively, using the membrane builder module of CHARMM-GUI web server (*Jo et al., 2008*; *Jo et al., 2009*). The systems were solvated using a TIP3P water model with a minimum margin of 20 Å between the protein and the edges of the periodic box in the z-axis. Chloride and sodium ions were added to reach a concentration of 150 mM and neutralize the total charge of the system. MD simulations were carried out using the Amber ff19SB (*Tian et al., 2020*) and Lipid 17 force field topologies with the parameters implemented in Amber 18 and AmberTools package (*Salomon-Ferrer et al., 2013*). The systems were energy minimized and equilibrated with NVT and NPT ensembles following the six-step preparation protocol recommended by CHARMM-GUI (*Lee et al., 2016*). Langevin thermostat algorithm was used to maintain the temperature at 310 K and the Monte Carlos barostat to set the pressure at 1.0 bar. All bonds involving hydrogens were constrained using the SHAKE algorithm. Each DWORF model was subjected to 4 µs MD simulation, while three independent replicates of the SERCA-DWORF complex were subjected to 1 µs MD simulations. For the MD analysis, MDAnalysis

python library (*Michaud-Agrawal et al., 2011*) was used to convert AMBER MD trajectories and coordinates to GROMACS files. We calculated the backbone RMSD, the tilt angle relative to the membrane normal of the TMD and AH (*helanal* library), the secondary structure (*do_dssp*), helix kink angle, number of contacts (*mindist*) and selection of residues within 3.0 Å of a query structure (*select*) for MD trajectories using the built-in tools of GROMACS 5.1.4 (*Abraham et al., 2015*) and MDAnalysis. Local membrane property analysis was performed for both systems with the *g_lomepro* package (*Gapsys et al., 2013*) implemented in GROMACS. The frames of each simulation were subjected to a backbone RMSD clustering analysis using a cutoff value of 2.0 and 2.5 Å to retrieve the most representative structures of DWORF and SERCA-DWORF, respectively. The plots were constructed with Gnuplot v5.2 (http://www.gnuplot.info) and figures were generated with PyMOL v1.7.

## Acknowledgements

This research was supported in part through computational resources and services provided by Advanced Research Computing at the University of Michigan, Ann Arbor, and Compute Canada (http://www.computecanada.ca). Funding: This work was supported in part by grants from the National Institutes of Health (R01HL092321 and R01HL143816 to HSY and SLR; R01GM120142 and R01HL148068 to LMEF; and R01HL130231 to AVZ), from the Heart and Stroke Foundation of Canada (to HSY), and the Natural Sciences and Engineering Research Council of Canada (RGPIN-2016-06478 to MJL).

## Additional information

### Competing interests

M Joanne Lemieux: Reviewing editor, *eLife*. The other authors declare that no competing interests exist.

### Funding

| Funder | Grant reference number | Author |
|---|---|---|
| National Institutes of Health | R01HL092321 | Seth L Robia<br>Howard S Young |
| National Institutes of Health | R01HL143816 | Seth L Robia<br>Howard S Young |
| National Institutes of Health | R01GM120142 | L Michel Espinoza-Fonseca |
| National Institutes of Health | R01HL148068 | L Michel Espinoza-Fonseca |
| National Institutes of Health | R01HL130231 | Aleksey V Zima |
| Natural Sciences and Engineering Research Council of Canada | RGPIN-2016-06478 | M Joanne Lemieux |
| Heart and Stroke Foundation of Canada | | Howard S Young |

The funders had no role in study design, data collection and interpretation, or the decision to submit the work for publication.

### Author contributions

M'Lynn E Fisher, Conceptualization, Data curation, Formal analysis, Investigation, Methodology; Elisa Bovo, Rodrigo Aguayo-Ortiz, Formal analysis, Investigation, Methodology, Writing - review and editing; Ellen E Cho, Formal analysis, Investigation; Marsha P Pribadi, Michael P Dalton, Formal analysis, Investigation, Methodology; Nishadh Rathod, Investigation, Methodology; M Joanne Lemieux, Resources, Methodology, Writing - review and editing; L Michel Espinoza-Fonseca, Formal analysis, Funding acquisition, Investigation, Methodology, Writing - review and editing; Seth L Robia, Conceptualization, Data curation, Formal analysis, Funding acquisition, Methodology, Writing - review

and editing; Aleksey V Zima, Conceptualization, Formal analysis, Funding acquisition, Investigation, Methodology, Writing - review and editing; Howard S Young, Conceptualization, Resources, Data curation, Formal analysis, Supervision, Funding acquisition, Investigation, Visualization, Methodology, Writing - original draft, Project administration, Writing - review and editing

### Author ORCIDs
Michael P Dalton (ID) http://orcid.org/0000-0001-5296-5099
Seth L Robia (ID) https://orcid.org/0000-0002-1193-9510
Howard S Young (ID) https://orcid.org/0000-0002-5990-8422

### Decision letter and Author response
Decision letter https://doi.org/10.7554/eLife.65545.sa1
Author response https://doi.org/10.7554/eLife.65545.sa2

## Additional files

### Supplementary files
• Transparent reporting form

### Data availability
All data generated or analyzed during this study are included in the manuscript.

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
