## [Decision Letter]

**Acceptance summary:**

The authors found that the peptide, dwarf open reading frame (DWORF), is a direct activator of the sarcoplasmic reticulum (SR) calcium pump (SERCA), which is responsible for maintaining high calcium within the SR. DWORF is the first known peptide activator of SERCA. Furthermore, they show that DWORF has a high affinity of SERCA in the presence of calcium.

**Decision letter after peer review:**

[Editors’ note: the authors submitted for reconsideration following the decision after peer review. What follows is the decision letter after the first round of review.]

Thank you for submitting your work entitled "Dwarf open reading frame (DWORF) is a direct activator of the sarcoplasmic reticulum calcium pump SERCA" for consideration by *eLife*. Your article has been reviewed by 3 peer reviewers, and the evaluation has been overseen by a Reviewing Editor and a Senior Editor. The following individual involved in review of your submission has agreed to reveal their identity: Amreen Mughal (Reviewer #1).

We are sorry to say that, after extensive consultation with the reviewers, we have decided that your work will not be considered further for publication by *eLife*.

It was felt that the conclusions were not supported by the data. Also previous studies including by one of the current co-authors provided clear evidence that DWORF competes with phospholamban, contrary to the current study. The general interest is also unclear for the readership of *eLife* without some understanding of physiological regulation of DWORF's interaction with SERCA.

*Reviewer #1:*

In this manuscript, Fished and Bravo et al. show that DWORF can act as a direct activator of SERCA. Building on seminal earlier work from Nelson et al. (2016), they proposed that DWORF increases SERCA dependent calcium reuptake. The findings are interesting, but in the absence of in vivo experiments, have limited physiological relevance.

The authors raised an important question, namely why an additional DWORF dependent mechanism is necessary for reversing PLN inhibition, but they have not yet answered this question.

Comments for the authors:

Introduction:

1. Authors did not describe Sarcolipin in the introduction. As they performed experiments with Sarcolipin as well, it is important that authors introduce it and provide information to the readers: why those experiments are important for this manuscript.

2. Authors raised an important question "This raised the question, why is an additional means of reversing PLN inhibition necessary?" But they did not answer this question in the manuscript. It will be worth if authors include some relevant discussion based on their results to show why having DWORF dependent SERCA activation mechanism is important.

Results:

3. Figure 4A: There is a baseline shift in Ca^2+^[ER] in DWORF group compared to control and PLN. This shift could cause an increase in ER Ca^2+^ concentration, but absolute change looks similar to other groups. Authors should clarify these results.

4. Figure 4: Please include representative cell images showing changes in Ca^2+^ concentration in different conditions.

5. All the experiments are performed in the absence of Phospholamban. However, phospholamban will be present in the physiological conditions, hence it is challenging to understand the physiological relevance of this work. There are no in vivo experiments performed to supports these results. Hence, authors should include this as a limitation.

*Reviewer #2:*

This work report the new finding that Dwarf open reading frame (DWORF) directly stimulates SERCA activity in vitro and in situ and it provides a molecular model for this action derived from FRET experiments and Molecular Dynamics simulations. The authors propose an interesting model for DWORF stimulation of SERCA, which awaits structural determination of the DWORF/SERCA2a complex: that the perturbation of the cytosolic side of the lipid bilayer may impact M1 and M2 movements near the Ca^2+^ access funnel and facilitate pump loading and concomitantly stimulate transport efficiency.

Strengths:

The combination of multiple techniques to demonstrate the direct interaction between DWORF and SERCA, as well as the clear stimulation of SERCA enzymatic activity (ATPase) and transport activity (Ca^2+^ uptake) in the presence and more importantly absence of phosopholamban convincingly validates the hypothesis that DWORF directly activates SERCA. The co-reconstitution of SERCA with DWORF clearly shows enzymatic increases to levels comparable to maxes reported in the presence of high phospholamban and the known SERCA agonist CDN1163. The functional measurements of Ca^2+^ transport and ER loading in the stable SERCA2a HEK-293 cell line is an excellent complement to the enzymology and provides additional confidence for direct DWORF activation. Clearly, this observation expands our knowledge of how SERCA activity is cellularly regulated and is likely an important component to our understanding of excitation-contraction coupling.

Weaknesses:

There are no apparent weaknesses with experimental results or the stated interpretation of these data. However, the discussion of the proposed model of DWORF stimulation generated by the modeling leaves a few unaddressed questions, the most significant being the following. The molecular modeling suggests that DWORF, like PLN binds in the inhibitory groove formed by M2, M6, and M9. Indeed, the proposed mechanism of PLN inhibition of SERCA is by competing for the enzyme conformation which binds Ca^2+^; specifically, the binding of Ca^2+^ causes a shift in M2 that closes the inhibitory groove such that the binding of Ca^2+^ and PLN are mutually exclusive. It is then puzzling how Ca^2+^-binding facilitates DWORF binding, if indeed it is within the same inhibitory groove.

Comments for the authors:

1) The only grammatical errors identified is in the Methods on page 12, line 6. "Data was fit with…." Should be "Data were fit with…."

2) The authors talk about structural differences between PLN and DWORF and the reader can make their own inferences as to how these minor differences might explain how DWORF binding is stabilized by Ca^2+^ binding, where PLN is destabilized. Nevertheless, this is something the authors may consider discussing a bit further as they certainly must have pondered the dichotomy. Along these lines, a few of the authors published a recent paper suggesting that SERCA dimerization plays a role in pump stimulation (Biophysical J. 119(7), 2020). It may be worth discussing how DWORF may contribute (or not) to the quaternary structure of SERCA and its physiologic function.

3) A few years ago a couple of the current authors published similar work on DWORF stimulation of SERCA (Makarewich et al., *eLife* 2018;7:e38319). In that report, they were pretty certain the DWORF mode of action was via competition with phospholamban. Indeed, the reviewers asked them to address the possibility that DWORF might directly modulate SERCA activity and they performed further experiments and concluded that DWORF only worked via PLN competition. Here is Makarewich et al. rebuttal to the reviewer's inquiry, "We have performed suggested experiments 'a' and 'c' and have included these results as Figure 2—figure supplement 3. Consistent with our previous publication on DWORF (Nelson et al., 2016), we found that co-expression of DWORF with unregulated SERCA did not lead to a direct or additive stimulation of SERCA activity (Figure 2—figure supplement 3B and C). Instead, we found that the ability of DWORF to simulate SERCA activity lies in its capacity to compete PLN off of SERCA and reduce the inhibitory effects of PLN on SERCA. This competitive displacement of PLN from SERCA can be very clearly seen in immunoprecipitation experiments shown in Figure 2—figure supplement 3A." Thus, this apparent mechanistic reversal deserves some explanation beyond "Compared to previous reports (12),…" stated on page 3. Specifically, in the 2018 *eLife* paper, Figure 2, supplement Figure 3B and 3C (purple), the direct stimulation of SERCA by DWORF is not apparent. What was missing in those previous experiments that prevented the observation that seems clear in the current manuscript? This information may be useful to other investigators studying SERCA regulation.

*Reviewer #3:*

In this study, Fisher et al. investigated the interaction of the sarco-endoplasmic reticulum calcium pump (SERCA) and the SERCA-binding protein small dwarf open reading frame (DWORF). Previous studies have shown that DWORF does not directly activate the SERCA pump and does not change the apparent affinity of SERCA for Ca+2. The current consensus in the field is that DWORF regulates the relative affinity of SERCA for Ca^2+^ by relieving the inhibition of PLN. In contrast to previous studies, the authors claim that DWORF directly regulates SERCA activity and calcium reuptake that is independent of its known role in relieving the inhibition of PLN. However, this major claim is not supported by the data presented in the manuscript.

Comments for the authors:

1. In Figure 2b it is shown that DWORF directly increases SERCA activity. Is this effect dose depended? Does the molar ratio of DWORF/SERCA affects the Vmax? It is also important to show the regulation of SERCA activity by DWORF in the presence and absence of PLN, phosphorylated PLN and SLN.

2. Related to the previous comment, there is a leftward shift of the Ca^2+^ affinity curve in the presence of DWORF (Figure 2b), suggesting that the affinity for Ca^2+^ is increased. Is this significant? How do the authors explain this effect?

3. In Figure 2a the data shows that PLN increases the Vmax of SERCA. This is puzzling as PLN is known to regulate the affinity os SERCA for Ca^2+^ but does not affect the Vmax. Is there an explanation for this observation? Also, the dose response curve and error bars should be shown on the graph.

4. The dose response curve of the small molecule CDN1163 should be shown in Figure 2C.

5. The authors used the skeletal muscle SERCA isoform SERCA1b in the reconstituted vesicles system and cardiac isoform, Serca2a, in the cell-based experiments. Are there any particular reasons why different isoforms were used in different experiments? Does DWORF regulate the activity of all SERCA isoforms?

6. Figure 3. How do the authors explain the discrepancies between the findings of the current study and the previous studies? For example, using a similar cellular model of DWORF overexpression (Cos7 cells co-transfected with SERCA2a) it has been reported that cells expressing SERCA2a and DWORF alone do not exhibit enhanced SERCA activity indicating that DWORF itself does not biochemically activate SERCA. Additional experiments are needed to clary this discrepancy. What is the effect of DWORF on SERCA2a in the presence of PLN and/or SLN in the cellular model used here? Is the Ca^2+^ uptake kinetics affected by the stoichiometry of the overexpressed proteins? As the levels of the overexpressed proteins are not titrated, it is likely that these observations represent artifacts associated with protein overexpression.

7. Figure 4a. The calcium transient frequency is significantly increased in the presence of DWORF. This suggest an increased RYR activity in the presence of DOWRF. Is there an indirect/direct effect of the RyR2 overexpression on the observed SERCA Ca^2+^ kinetics? Is RyR2 necessary for the regulation of SERCA by DWORF?

8. Figure 4b. Ideally the Ca^2+^ reuptake kinetics, eg Tau, should be measured under electrically stimulated conditions as the frequency of the Ca^2+^ release will affected the kinetics. It is likely that the increase of Tau by DWORF reflect the increase of the oscillation frequency.

---

## [Author Response]

[Editors’ note: The authors appealed the original decision. What follows is the authors’ response to the first round of review.]

Reviewer #1:[…] Comments for the authors:Introduction:1. Authors did not describe Sarcolipin in the introduction. As they performed experiments with Sarcolipin as well, it is important that authors introduce it and provide information to the readers: why those experiments are important for this manuscript.

In our manuscript, we presented previously published data for SERCA-SLN and SEERCA-PLN for comparison to SERCA-DWORF. However, after further consideration we decided the comparison to SERCA-SLN is not physiologically relevant and it has been removed (Figures 1 and 2 have been modified).

2. Authors raised an important question "This raised the question, why is an additional means of reversing PLN inhibition necessary?" But they did not answer this question in the manuscript. It will be worth if authors include some relevant discussion based on their results to show why having DWORF dependent SERCA activation mechanism is important.

The answer to this question is that DWORF is more than just an additional means of reversing PLN inhibition. DWORF is a direct activator of SERCA, which increases the dynamic physiological range of calcium that can be mobilized in response to demand for cardiac output. This question appears in the Introduction (on page 3 of our manuscript) and we have modified this paragraph to provide an immediate answer and preview of our results. The statement now reads: “This raised the question, why is an additional means of reversing PLN inhibition necessary? PLN inhibition can be reversed by phosphorylation (13, 14), elevated calcium concentrations, and we can now add DWORF as another level of redundancy in reversing SERCA inhibition by PLN. In the present study, we show that DWORF is a direct activator of SERCA activity, in addition to its role in displacing PLN from the inhibitory groove of SERCA.”

Results:3. Figure 4A: There is a baseline shift in Ca^2+^[ER] in DWORF group compared to control and PLN. This shift could cause an increase in ER Ca^2+^ concentration, but absolute change looks similar to other groups. Authors should clarify these results.

We will be happy to clarify these results. The baseline shift is caused by an increase in ER calcium concentration, which is due to increased SERCA activity with DWORF expression. This can be seen in the upper panel of Figure 4B. DWORF causes both a baseline shift (increase in ER calcium load) and an absolute change (increase in the amplitude of calcium release from the ER). Changes in these two parameters are directly related to changes in SERCA function and they underscore the functional significance of the novel mechanisms described in our manuscript. We have clarified the description of these results to indicate that ER calcium load increases with DWORF co-expression (page 4). The text now reads:

“Coexpression of DWORF resulted in increased ER calcium load (Figure 4B; [Ca^2+^]_ER_) and increased frequency (Figure 4B; Tau(s)) and amplitude of spontaneous calcium waves, suggesting a significantly faster calcium reuptake due to increased SERCA activity.”

4. Figure 4: Please include representative cell images showing changes in Ca^2+^ concentration in different conditions.

We have added representative cell image line scans showing changes in calcium concentration as Supplemental Figure 1. Further experimental details for these methods, as well as example cell images and line scans, have been previously published and can be found in Bovo et al., Am J Physiol Heart Circ Physiol (2019) 316:H1323-H1331.

5. All the experiments are performed in the absence of Phospholamban. However, phospholamban will be present in the physiological conditions, hence it is challenging to understand the physiological relevance of this work. There are no in vivo experiments performed to supports these results. Hence, authors should include this as a limitation.

The in vivo experiments have been done by others for DWORF in the presence of phospholamban (1,2). The original paper describing DWORF looked at WT, DWORF overexpression, and DWORF knockout mice (2). All mouse models had normal phospholamban expression levels. Peak systolic calcium transient amplitude and SR calcium load were significantly increased in the DWORF overexpression adult cardiomyocytes. These data are completely consistent with the novel observations presented in our manuscript.

As requested by the reviewer, we have elaborated on these points in the Discussion section of our revised manuscript (page 7). The following statements have been added:

“Previous studies of DWORF overexpression in mouse models with normal PLN expression level demonstrated that peak systolic calcium transient amplitude and SR calcium load were significantly increased with DWORF overexpression (11, 12). While these findings were attributed to DWORF displacement of PLN and relief of SERCA inhibition, these findings are also consistent with a dual mechanism that includes direct activation of SERCA by DWORF.”

Reviewer #2:[…] There are no apparent weaknesses with experimental results or the stated interpretation of these data. However, the discussion of the proposed model of DWORF stimulation generated by the modeling leaves a few unaddressed questions, the most significant being the following. The molecular modeling suggests that DWORF, like PLN binds in the inhibitory groove formed by M2, M6, and M9. Indeed, the proposed mechanism of PLN inhibition of SERCA is by competing for the enzyme conformation which binds Ca^2+^; specifically, the binding of Ca^2+^ causes a shift in M2 that closes the inhibitory groove such that the binding of Ca^2+^ and PLN are mutually exclusive. It is then puzzling how Ca^2+^-binding facilitates DWORF binding, if indeed it is within the same inhibitory groove.

The reviewer’s assessment that “the binding of Ca^2+^ and PLN are mutually exclusive” is not supported by current evidence. Early models for the SERCA-PLN interaction suggested that calcium relieves SERCA inhibition by dissociating the complex. However, there is now extensive evidence indicating that PLN remains bound to SERCA in the presence of calcium (e.g. (3-6)). In addition, the crystal structure of the SERCA-PLN complex clearly shows that SERCA is in an E1-like state similar to the calcium-bound state of SERCA (7-9). Importantly, the FRET data shown in Figure 5 of our manuscript clearly shows that PLN remains associated with SERCA in the presence of calcium, though the binding affinity is lower.

We have clarified this point in the Discussion section of our manuscript (page 8). The statement now reads:

“Previous studies have shown that PLN remains associated with SERCA even in the presence of calcium (e.g. (24, 42-44)). […] In contrast, calcium increased the apparent affinity of DWORF for SERCA (Figure 5B and C), revealing that DWORF prefers to interact with conformations of SERCA that prevail at high calcium.”

Comments for the authors:1) The only grammatical errors identified is in the Methods on page 12, line 6. "Data was fit with…." Should be "Data were fit with…."

This has been corrected.

2) The authors talk about structural differences between PLN and DWORF and the reader can make their own inferences as to how these minor differences might explain how DWORF binding is stabilized by Ca^2+^ binding, where PLN is destabilized. Nevertheless, this is something the authors may consider discussing a bit further as they certainly must have pondered the dichotomy. Along these lines, a few of the authors published a recent paper suggesting that SERCA dimerization plays a role in pump stimulation (Biophysical J. 119(7), 2020). It may be worth discussing how DWORF may contribute (or not) to the quaternary structure of SERCA and its physiologic function.

The reviewer raises a very interesting question. We do not have direct experimental evidence of a relationship between DWORF/PLN regulation and SERCA dimerization; as far as we know these are distinct structure/function mechanisms. However, both dimerization and DWORF/PLN modulate specific steps in the enzymatic cycle so there may be an undiscovered mechanistic connection. While this is beyond the scope of the present manuscript, a future detailed kinetic study may correlate SERCA dimer conformational coupling and SERCA regulation by DWORF/PLN.

3) A few years ago a couple of the current authors published similar work on DWORF stimulation of SERCA (Makarewich et al., eLife 2018;7:e38319). In that report, they were pretty certain the DWORF mode of action was via competition with phospholamban. Indeed, the reviewers asked them to address the possibility that DWORF might directly modulate SERCA activity and they performed further experiments and concluded that DWORF only worked via PLN competition. Here is Makarewich et al. rebuttal to the reviewer's inquiry, "We have performed suggested experiments 'a' and 'c' and have included these results as Figure 2—figure supplement 3. Consistent with our previous publication on DWORF (Nelson et al., 2016), we found that co-expression of DWORF with unregulated SERCA did not lead to a direct or additive stimulation of SERCA activity (Figure 2—figure supplement 3B and C). Instead, we found that the ability of DWORF to simulate SERCA activity lies in its capacity to compete PLN off of SERCA and reduce the inhibitory effects of PLN on SERCA. This competitive displacement of PLN from SERCA can be very clearly seen in immunoprecipitation experiments shown in Figure 2—figure supplement 3A." Thus, this apparent mechanistic reversal deserves some explanation beyond "Compared to previous reports (12),…" stated on page 3. Specifically, in the 2018 eLife paper, Figure 2, supplement Figure 3B and 3C (purple), the direct stimulation of SERCA by DWORF is not apparent. What was missing in those previous experiments that prevented the observation that seems clear in the current manuscript? This information may be useful to other investigators studying SERCA regulation.

We thank the reviewer for raising this concern, as the answer to this question is quite simple. The previous studies of DWORF, including the original paper (2) and the mentioned *eLife* paper (1), focused on the apparent affinity of SERCA for calcium (KCa), which is not significantly changed by DWORF. They did not examine the maximal activity (Vmax) of SERCA in these studies, which is the parameter modulated by DWORF. It is difficult to measure Vmax from mouse heart homogenates, but this parameter is accessible in the well-defined biochemical preparations used in the present study. In summary, we do not consider our present results to be in conflict with the previous papers (1,2), as we did not observe an effect of DWORF on the KCa of SERCA. Our results provide a new facet of SERCA regulation by DWORF and build on the important contributions of previous studies (1,2).

To clarify this point, we have added statements to the Results section of our manuscript (page 3). The statements read:

“Previous studies of DWORF function focused on the apparent calcium affinity and not the maximal activity of SERCA in the presence of DWORF (11, 12).” And “Previous studies have shown that DWORF relieves SERCA inhibition by displacing PLN from the inhibitory groove of SERCA (11, 12). Our data are consistent with this observation, and we add a new facet of DWORF function as a direct activator of SERCA even in the absence of PLN.”

Note that during the review of our manuscript by *eLife*, a publication appeared in the Journal of Biological Chemistry (Li, Yuen, Stroik, Kleinboehl, Cornea, and Thomas). The authors claim to show that DWORF activates SERCA by altering the calcium affinity of SERCA. These data are in conflict with previous studies (1,2), as well as the data in our manuscript. We have addressed this inconsistency in the revised version of our manuscript. We have added the following statement to the Results section of our manuscript (page 3):

“A recent study claimed to show that DWORF increases the apparent calcium affinity of SERCA (15), though our current data (Figure 2) and previous studies by others (11, 12) do not support their conclusion.”

The publication of the Li et al. study further increases the urgency for dissemination of our data and manuscript, which clearly show that DWORF does not alter the KCa of SERCA and instead it increases the Vmax of SERCA. Note that the manuscript deposited in BioRxiv by the Veglia lab (https://www.biorxiv.org/content/10.1101/2021.05.05.442831v1) also shows that DWORF increases the Vmax of SERCA.

Reviewer #3:In this study, Fisher et al. investigated the interaction of the sarco-endoplasmic reticulum calcium pump (SERCA) and the SERCA-binding protein small dwarf open reading frame (DWORF). Previous studies have shown that DWORF does not directly activate the SERCA pump and does not change the apparent affinity of SERCA for Ca+2. The current consensus in the field is that DWORF regulates the relative affinity of SERCA for Ca^2+^ by relieving the inhibition of PLN. In contrast to previous studies, the authors claim that DWORF directly regulates SERCA activity and calcium reuptake that is independent of its known role in relieving the inhibition of PLN. However, this major claim is not supported by the data presented in the manuscript.

We would like to point out that our data are completely consistent with the current model that DWORF competes with PLN for SERCA. The previous studies of the DWORF functional effect on SERCA only examined the effect on the KCa of SERCA. Our results agree with their observation; we also see no effect on KCa. Importantly, those previous studies did not look at the maximal activity (Vmax) of SERCA, thus our discovery of a direct effect on Vmax by DWORF are not in conflict with previous reports (1,2). In addition, the in-cell studies are unique and have not been performed by others to the best of our knowledge. The in-cell studies are also completely consistent with previous reports (1,2).

This same concern was raised by reviewer #2, please refer to question 3 above and our response. Our results confirm the previous observation that DWORF does not alter SERCA calcium affinity (2,3) and we have added two new mechanistic insights: (i) DWORF and PLN bind to SERCA in a calcium-dependent manner and (ii) DWORF directly increases SERCA maximal activity (Vmax). In support of these proposed novel mechanisms, we provide biochemical measurements of SERCA ATPase activity and unique, quantitative in-cell measurements of SERCA calcium transport kinetics.

Comments for the authors:1. In Figure 2b it is shown that DWORF directly increases SERCA activity. Is this effect dose depended? Does the molar ratio of DWORF/SERCA affects the Vmax? It is also important to show the regulation of SERCA activity by DWORF in the presence and absence of PLN, phosphorylated PLN and SLN.

Indeed, we have done a dose response of DWORF in the presence of SERCA. At equimolar ratios of SERCA-DWORF, DWORF directly activates SERCA as presented. Based on the available literature, this condition best represents the physiological situation. Under extreme, non-physiological conditions of excess DWORF, we have observed modest SERCA inhibition manifested as a decrease in calcium affinity and a decrease in maximal activity. We have chosen not to include this latter data as it is certainly non-physiological, and it will confuse readers. The revised manuscript emphasizes the importance of controlling this parameter. The following statement has been added to the Methods section (page 10):

“To ensure the proper ratio of ~2 DWORF per SERCA in the reconstituted proteoliposomes, quantitative SDS-PAGE was used with known quantities of SERCA and DWORF proteins as standards (17, 18, 50).”

For the second point, our goal was to elucidate a direct effect of DWORF on SERCA and our results clearly support this novel mechanism. It is generally accepted that DWORF displaces PLN and our goal was not to reiterate the conclusions of previous reports (1,2).

2. Related to the previous comment, there is a leftward shift of the Ca^2+^ affinity curve in the presence of DWORF (Figure 2b), suggesting that the affinity for Ca^2+^ is increased. Is this significant? How do the authors explain this effect?

There is a small rightward shift (not a leftward shift) of the calcium affinity curve in the presence of DWORF (in Figure 2b), but this is not statistically significant.

3. In Figure 2a the data shows that PLN increases the Vmax of SERCA. This is puzzling as PLN is known to regulate the affinity os SERCA for Ca^2+^ but does not affect the Vmax. Is there an explanation for this observation? Also, the dose response curve and error bars should be shown on the graph.

SERCA and PLN are present together in cardiac SR membranes at molar ratios ranging from 1:1 to 1:4. At the higher molar ratios (1:4), PLN inhibits SERCA at low calcium concentrations (effect on KCa) and it increases the maximal activity of SERCA at higher calcium concentrations. The Young lab has previously studied this effect (10-15). Figure 2a showed fitted curves from previously published data and it was presented for comparison with the new SERCA-DWORF data. In retrospect, the effect of PLN on SERCA at different molar ratios is not important for this study and is confusing for the reader. We have modified the figure to show the effect of PLN on SERCA at an equimolar 1:1 ratio, which will better compare to the data for equimolar ratio of DWORF and SERCA.

4. The dose response curve of the small molecule CDN1163 should be shown in Figure 2C.

The CDN1163 compound has been thoroughly studied and published by others (16). We used this compound in our system as a control for comparison with the activation of SERCA by DWORF.

5. The authors used the skeletal muscle SERCA isoform SERCA1b in the reconstituted vesicles system and cardiac isoform, Serca2a, in the cell-based experiments. Are there any particular reasons why different isoforms were used in different experiments? Does DWORF regulate the activity of all SERCA isoforms?

The reconstitution of SERCA for quantitative ATPase measurements requires a lot of protein, which is readily available from rabbit skeletal muscle. Because it expresses high levels of the SERCA1a isoform, it is the standard source of enzyme in the field. The two isoforms (SERCA1a and 2a) are 84% identical, and their molecular structures have been determined to be identical (17). We have previously shown that PLN regulates SERCA1a and SERCA2a in a similar manner (18). Most important, DWORF has been shown to co-immunoprecipitate with all SERCA isoforms (2).

6. Figure 3. How do the authors explain the discrepancies between the findings of the current study and the previous studies? For example, using a similar cellular model of DWORF overexpression (Cos7 cells co-transfected with SERCA2a) it has been reported that cells expressing SERCA2a and DWORF alone do not exhibit enhanced SERCA activity indicating that DWORF itself does not biochemically activate SERCA. Additional experiments are needed to clary this discrepancy. What is the effect of DWORF on SERCA2a in the presence of PLN and/or SLN in the cellular model used here? Is the Ca^2+^ uptake kinetics affected by the stoichiometry of the overexpressed proteins? As the levels of the overexpressed proteins are not titrated, it is likely that these observations represent artifacts associated with protein overexpression.

Please see our responses to similar questions above. In our revision, we have explained more clearly that there is no discrepancy. In a previous study, COS7 cells co-transfected with SERCA2a and DWORF were used for co-immunoprecipitation studies and measurement of the calcium affinity of SERCA (2). To quote this study “We found that coexpression of DWORF alone with SERCA2a did not change the apparent affinity of SERCA for Ca^2+^, but it relieved the inhibition by PLN in a dose-dependent manner (Figure S15).” Thus, these studies focused on the KCa of SERCA and not the Vmax. Our results are completely consistent with these prior studies by the Olson laboratory (2).

7. Figure 4a. The calcium transient frequency is significantly increased in the presence of DWORF. This suggest an increased RYR activity in the presence of DOWRF. Is there an indirect/direct effect of the RyR2 overexpression on the observed SERCA Ca^2+^ kinetics? Is RyR2 necessary for the regulation of SERCA by DWORF?

Our data shows that DWORF increases the ER calcium content, which is dependent on the presence of SERCA. As the ER calcium content increases, the frequency of spontaneous calcium release mediated by RyR2 also increases. A similar effect is observed in cardiomyocytes. Calcium transient (calcium wave) frequency is increased in cells expressing DWORF (Figure 4C). This effect is mediated by increased SERCA activity and the resulting increase in ER calcium load. Indeed, it is well known that an increase in SR calcium load during SERCA activation (adrenergic activation) increases the frequency of spontaneous calcium transients in myocytes.

We can rule out an effect of DWORF on RyR, since it has been shown by many labs (including (19)) that RyR activation would only transiently increase calcium transient frequency. As SR calcium gets depleted during RyR activation, calcium transients would be reduced to control levels. Sustained increases in spontaneous calcium transient frequency can only be achieved by increased SERCA activity.

Moreover, in vitro ATPase measurements show that RyR is not required for SERCA activation by DWORF. We express RyR for the live cell calcium uptake assay as a way to manipulate ER calcium load (by producing spontaneous CICR or by activating/inhibiting RyR), so SERCA function can be studied throughout a wide range of ER calcium loads.

8. Figure 4b. Ideally the Ca^2+^ reuptake kinetics, eg Tau, should be measured under electrically stimulated conditions as the frequency of the Ca^2+^ release will affected the kinetics. It is likely that the increase of Tau by DWORF reflect the increase of the oscillation frequency.

The HEK cells expressing SERCA, RyR, DWORF and PLN are a useful incell cardiomimetic system; however, it is not possible to do electrical stimulation of HEK cells. Calcium release frequency only affects uptake kinetics insofar as it may affect the ER calcium load, a parameter that is carefully quantified here. An important aspect of our in-cell experiments is the calibration of the fluorescent ER calcium indicator at the end of each experiment. ER content calibration allows us to specifically determine uptake rate as a function of ER calcium load. We also control the cytosolic calcium concentration (buffered to 100 nM), and we quantify ER leak rate independent of SERCA and RyR. Our unique approach yields quantitative measurements of maximal uptake rate and maximal ER calcium load as intrinsic features of SERCA function.

References:

1. Makarewich, C. A., Munir, A. Z., Schiattarella, G. G., Bezprozvannaya, S., Raguimova, O. N., Cho, E. E., Vidal, A. H., Robia, S. L., Bassel-Duby, R., and Olson, E. N. (2018) The DWORF micropeptide enhances contractility and prevents heart failure in a mouse model of dilated cardiomyopathy. e*Life* 7

2. Nelson, B. R., Makarewich, C. A., Anderson, D. M., Winders, B. R., Troupes, C. D., Wu, F., Reese,

A. L., McAnally, J. R., Chen, X., Kavalali, E. T., Cannon, S. C., Houser, S. R., Bassel-Duby, R., and Olson, E. N. (2016) A peptide encoded by a transcript annotated as long noncoding RNA enhances SERCA activity in muscle. Science 351, 271-275

3. Bidwell, P., Blackwell, D. J., Hou, Z., Zima, A. V., and Robia, S. L. (2011) Phospholamban binds with differential affinity to calcium pump conformers. J Biol Chem 286, 35044-35050

4. Dong, X., and Thomas, D. D. (2014) Time-resolved FRET reveals the structural mechanism of SERCA-PLB regulation. Biochem Biophys Res Commun 449, 196-201

5. Mueller, B., Karim, C. B., Negrashov, I. V., Kutchai, H., and Thomas, D. D. (2004) Direct detection of phospholamban and sarcoplasmic reticulum Ca-ATPase interaction in membranes using fluorescence resonance energy transfer. Biochemistry 43, 8754-8765

6. Li, J., Bigelow, D. J., and Squier, T. C. (2004) Conformational changes within the cytosolic portion of phospholamban upon release of Ca-ATPase inhibition. Biochemistry 43, 3870-3879

7. Akin, B. L., Hurley, T. D., Chen, Z., and Jones, L. R. (2013) The structural basis for phospholamban inhibition of the calcium pump in sarcoplasmic reticulum. J Biol Chem 288, 30181-30191

8. Toyoshima, C., Iwasawa, S., Ogawa, H., Hirata, A., Tsueda, J., and Inesi, G. (2013) Crystal structures of the calcium pump and sarcolipin in the Mg^2+^-bound E1 state. Nature 495, 260-264

9. Winther, A. M., Bublitz, M., Karlsen, J. L., Moller, J. V., Hansen, J. B., Nissen, P., and BuchPedersen, M. J. (2013) The sarcolipin-bound calcium pump stabilizes calcium sites exposed to the cytoplasm. Nature 495, 265-269

10. Glaves, J. P., Primeau, J. O., Gorski, P. A., Espinoza-Fonseca, L. M., Lemieux, M. J., and Young, H.

S. (2020) Interaction of a Sarcolipin Pentamer and Monomer with the Sarcoplasmic Reticulum Calcium Pump, SERCA. Biophys J 118, 518-531

11. Glaves, J. P., Primeau, J. O., Espinoza-Fonseca, L. M., Lemieux, M. J., and Young, H. S. (2019) The Phospholamban Pentamer Alters Function of the Sarcoplasmic Reticulum Calcium Pump SERCA. Biophys J 116, 633-647

12. Ceholski, D. K., Trieber, C. A., and Young, H. S. (2012) Hydrophobic imbalance in the cytoplasmic domain of phospholamban is a determinant for lethal dilated cardiomyopathy. J Biol Chem 287, 16521-16529

13. Glaves, J. P., Trieber, C. A., Ceholski, D. K., Stokes, D. L., and Young, H. S. (2011) Phosphorylation and mutation of phospholamban alter physical interactions with the sarcoplasmic reticulum calcium pump. J Mol Biol 405, 707-723

14. Trieber, C. A., Afara, M., and Young, H. S. (2009) Effects of phospholamban transmembrane mutants on the calcium affinity, maximal activity, and cooperativity of the sarcoplasmic reticulum calcium pump. Biochemistry 48, 9287-9296

15. Trieber, C. A., Douglas, J. L., Afara, M., and Young, H. S. (2005) The effects of mutation on the regulatory properties of phospholamban in co-reconstituted membranes. Biochemistry 44, 3289-3297

16. Kang, S., Dahl, R., Hsieh, W., Shin, A., Zsebo, K. M., Buettner, C., Hajjar, R. J., and Lebeche, D.

(2016) Small Molecular Allosteric Activator of the Sarco/Endoplasmic Reticulum Ca^2+^-ATPase

(SERCA) Attenuates Diabetes and Metabolic Disorders. J Biol Chem 291, 5185-5198

17. Sitsel, A., De Raeymaecker, J., Drachmann, N. D., Derua, R., Smaardijk, S., Andersen, J. L., Vandecaetsbeek, I., Chen, J., De Maeyer, M., Waelkens, E., Olesen, C., Vangheluwe, P., and

Nissen, P. (2019) Structures of the heart specific SERCA2a Ca(2+)-ATPase. EMBO J 38

18. Gorski, P. A., Trieber, C. A., Ashrafi, G., and Young, H. S. (2015) Regulation of the sarcoplasmic reticulum calcium pump by divergent phospholamban isoforms in zebrafish. J Biol Chem 290, 6777-6788

19. Trafford, A. W., Diaz, M. E., Sibbring, G. C., and Eisner, D. A. (2000) Modulation of CICR has no maintained effect on systolic Ca^2+^: simultaneous measurements of sarcoplasmic reticulum and sarcolemmal Ca^2+^ fluxes in rat ventricular myocytes. J Physiol 522 Pt 2, 259-270